

# Turbulence Kinetic Energy dissipation rate: Assessment of radar models from comparisons between 1.3 GHz WPR and DataHawk UAV measurements

Hubert Luce[1], Laskhmi Kantha[2], Hiroyuki Hashiguchi[1], Dale Lawrence[2], Abhiram Doddi[2], Tyler Mixa[3], Masanori Yabuki[1]

[1]Research Institute for Sustainable Humanosphere, Kyoto University, Kyoto, 611-0011, Japan
[2]Smead Aerospace Engineering Sciences, University of Colorado, Boulder, CO, USA
[3]GATS Inc., Boulder, CO, USA

*Correspondence to*: Hubert Luce (luce@rish.kyoto-u.ac.jp)

**Abstract.** The WPR-LQ-7 is a UHF (1.3575 GHz) wind profiler radar used for routine measurements of the lower troposphere at Shigaraki MU observatory (34.85°N, 136.10°E, Japan) at a vertical resolution of 100 m and a time resolution of 10 min. Following studies carried out with the 46.5 MHz Middle and Upper atmosphere (MU) radar (Luce et al., 2018), we tested models used to estimate turbulence kinetic energy (TKE) dissipation rates $\varepsilon$ from the Doppler spectral width in the altitude range ~0.7 to 4.0 km ASL. For this purpose, we compared LQ-7-derived $\varepsilon$ by using processed data available on line (http://www.rish.kyoto-u.ac.jp/radar-group/blr/shigaraki/data/) with direct estimates of $\varepsilon$ ($\varepsilon_U$) from DataHawk UAVs. The statistical results reveal the same trends as reported by Luce et al. (2018) with the MU radar, namely: (1) The simple formulation based on dimensional analysis $\varepsilon_{Lout} = \sigma^3/L_{out}$, with $L_{out}$ ~70 m, provides the best statistical agreement with $\varepsilon_U$. (2) The model $\varepsilon_N$ predicting a $\sigma^2 N$ law ($N$ is Brunt-Vaïsälä frequency) for stably stratified conditions tends to overestimate for $\varepsilon_U < \sim 5\ 10^{-4}\ m^2 s^{-3}$ and to underestimate for $\varepsilon_U > \sim 5\ 10^{-4}\ m^2 s^{-3}$. We also tested a model $\varepsilon_S$ predicting a $\sigma^2 S$ law ($S$ is the vertical shear of horizontal wind) supposed to be valid for low Richardson numbers ($Ri = N^2/S^2$). From the case study of a turbulent layer produced by a Kelvin-Helmholtz instability, we found that $\varepsilon_S$ and $\varepsilon_{Lout}$ are both very consistent with $\varepsilon_U$, while $\varepsilon_N$ underestimates $\varepsilon_U$ in the core of the turbulent layer where $N$ is minimum. We also applied the Thorpe method from data collected from a nearly simultaneous radiosonde and tested an alternative interpretation of the Thorpe length in terms of the Corrsin scale defined for weakly stratified turbulence. A statistical analysis showed that $\varepsilon_S$ also provides better statistical agreement with $\varepsilon_U$ and is much less biased than $\varepsilon_N$. Combining estimates of $N$ and shear from DataHawk and radar data, respectively, a rough estimate of the Richardson number at a vertical resolution of 100 m ($Ri_{100}$) was obtained. We performed a statistical analysis on the $Ri$ dependence of the models. The main outcome is that $\varepsilon_S$ compares well with $\varepsilon_U$ for low $Ri_{100}'s$ ($Ri_{100} < \sim 1$) while $\varepsilon_N$ fails. $\varepsilon_{Lout}$ varies as $\varepsilon_S$ with $Ri_{100}$ so that $\varepsilon_{Lout}$ remains the best (and simplest) model in the absence of information on $Ri$. Also, $\sigma$ appears to vary as $Ri_{100}^{-1/2}$ when $Ri_{100} > \sim 0.4$ and shows a degree of dependence with $S_{100}$ otherwise.



## 1 Introduction

VHF Stratosphere-Troposphere radars and UHF wind profilers are commonly used to estimate turbulence kinetic energy (TKE) dissipation rate ε from Doppler spectral width (hereafter, noted $2\sigma_{obs}$) in the atmosphere (e.g.,Hocking, 1983, 1985, 1986, 1999; Fukao et al., 1994; Cohn, 1995, Kurosaki et al., 1996; Bertin et al., 1997, Delage et al., 1997; Naström and Eaton, 1997, Dole et al., 2001, Jacoby-Kaoly et al., 2002, Satheesan and Murthy, 2002; Naström and Eaton, 2005; Wilson et al., 2005, Kalapureddy et al., 2007; Singh et al., 2008; Dehghan and Hocking, 2011; Kantha and Hocking, 2011; Dehghan et al., 2014; Wilson et al., 2014; Hocking et al., 2016; Li et al., 2016; Kohma et al., 2019; Jaiswal et al., 2020; Chen et al., 2022). Several models have been proposed to relate $2\sigma_{obs}$ to ε. Some studies have accepted the validity of these models in order to perform statistical analyses of the turbulence characteristics in the tropo-stratosphere (e.g., Fukao et al. 1994; Kurosaki et al., 1996; Naström and Eaton, 1997; Kalapureddy et al., 2007; Fukao et al., 2011; Chen et al., 2022). Other studies have tested the consistency between models based on spectral width measurement and their consistency with other radar models based on echo power or radial wind velocity variance measurement (e.g. Cohn, 1995; Bertin et al., 1997; Delage et al., 1997; Satheesan and Krishna Murty, 2002; Singh et al., 2008). Yet others assessed the radar estimates from cross-comparisons with indirect estimates based on the Thorpe sorting method applied to potential temperature profiles measured by standard radiosondes (e.g. Clayson and Kantha, 2008; Kantha, 2010; Kantha and Hocking, 2011; Wilson et al., 2014; Li et al., 2016; Kohma et al., 2019, Jaiswal et al., 2020). However, attempts at validations from direct in-situ estimates of ε from velocity fluctuation measurements remain very rare. McCaffrey et al. (2017) compared ε estimates derived from a UHF wind profiler and those obtained from sonic anemometer energy spectra made at a 300-m altitude. Shaw and LeMone (2003) and Jacoby-Koaly et al. (2002) evaluated the performance of UHF wind profilers from in-situ ε aircraft and/or tower measurements mainly in the convective boundary layer. Dehghan et al. (2014) made ε comparisons between aircraft and the VHF (40.68 MHz) Harrow radar with mixed results.

In addition to being rare, the above-mentioned studies did not aim to test the same radar models. Luce et al. (2018), hereafter denoted L18, assessed different models from comparisons with direct estimates of ε from air speed fluctuation measurements made from highly sensitive Pitot sensors aboard DataHawk UAVs, and the VHF 46.5 MHz Middle and Upper atmosphere (MU) radar observations in the lower troposphere. One of the objectives of the present work is to show that the conclusions obtained from comparisons with the MU radar are also quantitatively valid for the WPR-LQ-7 (Imai et al., 2007), a UHF wind profiler routinely used at the Shigaraki MU Observatory. We also introduce and test another model expected to be valid for weakly stratified or strongly sheared conditions, i.e., low Richardson ($Ri$) numbers (Hunt et al., 1998; Basu et al., 2021; Basu and Holtslag, 2021) for which the static stability effects can be ignored. $Ri$ is defined as $N^2/S^2$ where $N^2 = g/\theta \, d\theta/dz$ is the Brunt-Väisälä frequency, g is the acceleration of gravity, $\theta$ is the potential temperature and $S$ is the vertical shear of horizontal wind vector.

Section 2 introduces the expressions for ε used in the present paper with a focus on the newly introduced model in radar studies. Section 3 briefly describes the WPR-LQ-7 and the methods used for the comparisons. Section 4 describes the results for two turbulent layers, one of which was clearly produced by a Kelvin-Helmholtz (shear flow) instability because of the observation of S-shaped structures specific to this instability in both time-height MU and WPR-LQ-7 echo power cross-sections. The results of comparisons of ε values obtained from the different models applied to the two radars, DataHawk measurements, and a simultaneous radiosonde using the Thorpe sorting method of vertical potential temperature profiles are described for the two layers. Section 5 shows statistics on the consistency between the estimates of ε from the different models and the DataHawks and describes the dependence of the models on Ri. Finally, conclusions are given in Section 6.

## 2. The radar models of ε

### 2.1 The models tested by L18





The different models and their conditions of application have already been described by L18. Here, the expressions are simply reintroduced. Assuming a vertically pointing radar beam, the first expression is:

$\quad \varepsilon_{Lout} = \sigma^3/L_{out}$ (1)

where $\sigma^2$ is an estimate of the variance $\langle w'^2 \rangle$ of the vertical wind fluctuations produced by turbulence. $L_{out}$ has the dimension of a length scale and represents the scale of energy containing turbulent eddies if $\sigma^2$ is an unbiased estimate of $\langle w'^2 \rangle$. In practice, $\sigma^2$ is obtained after removing the non-turbulent contributions from $\sigma_{obs}^2$ (e.g. Hocking, 1983; Naström, 1997; Hocking, 2016 and references therein). The practical method used in the present work is described in the Appendix of L18. The dissipation rate is expected to vary as $\sigma^3$ if $\sigma$ and $L_{out}$ are independent or when the typical scale of turbulent eddies exceed

$\quad$ the dimensions of the radar volume so that $L_{out}$ would mainly be function of these dimensions (e.g., Frisch and Clifford, 1974; Labitt, 1979; Doviak and Zrnic', 1984; White et al. 1999).

The second expression (e.g. Hocking, 1983; 1999; Hocking et al., 2016) is:

$\quad \varepsilon_N = C_N \sigma^2 N$ (2)

$\quad$ where $C_N$ is a constant ($= 0.5 \pm 0.25$) according to Hocking et al. (2016). Eq. (2) is expected to be applicable to turbulence under stably stratified conditions ($N^2 > 0$). Following pioneering works (e.g. Weinstock, 1978), the inertial subrange is assumed to be limited at large scales by a N-dependent scale since the largest turbulent eddies are expected to be affected by the stable stratification. By using the buoyancy scale expressed as $L_B = \sigma/N$, (2) is equivalent to $\varepsilon_N = C_N \sigma^3/L_B$.

The results of comparisons reported by L18 with DataHawk-derived $\varepsilon$ showed that (1) provides the best overall statistical

$\quad$ comparisons for $L_{out} \sim 50 - 70$ m. The analysis of the comparisons results with $\varepsilon_N$ suggested that N is not a key parameter since the quality of the comparisons appeared to be independent of N.

## 2.2 The model for strongly sheared or/and weakly stratified flows

Although it seems that the conditions of strong shear and weak stratification have not received much attention in the radar community, several studies showed that $\varepsilon$ can be written as (Hunt et al., 1988; Schumann and Gerz, 1995):

$\quad \varepsilon_S = C_S \sigma^2 S$ (3)

Eq. (3) is equivalent to $\varepsilon_S = C_S \sigma^3/L_H$ where $L_H = \sigma/S$ is the Hunt length scale. The Eq. (3) can be interpreted as the fact that turbulent eddies are first stretched by shear before being affected by stratification in strongly sheared or weakly stratified flows. This concept was discussed by Hocking and Hamza (1997) but they did not mention the Hunt scale and did not go further into it. Hunt et al. (1988) suggested that Eq. (3) can be valid up to Ri$\sim$0.5. Schumann and Gerz (1995) even proposed

$\quad$ up to Ri $\sim$1 from Large Eddy Simulations. Hunt et al. (1988) proposed $C_S \approx 0.45$ for neutral stationary boundary layers. Kaltenbach et al. (1994) found $0.54 < C_S < 0.62$ from Large Eddy Simulations. From a simplified TKE budget equation for a homogeneous shear layer in steady state, i.e., $\varepsilon = P - B$ where $P$ is the shear production and $B$ the buoyancy destruction term, and using similarity theory, Basu and Holtslag (2021) re-evaluated the constant $C_S$ and provided a generalization of Eq. (3):

$\quad \varepsilon_s' = C_S \left( \frac{1 - R_f}{Ri} \right)^{1/2} \sigma^2 N = C_S (1 - R_f)^{1/2} \sigma^2 S = C_s'(Rf) \sigma^2 S$ (4)

With $C_S = 0.63$. $R_f$ is the flux Richardson number. It is related to the turbulent Prandtl number Pr by $R_f = Ri/Pr$. Basu et al. (2021) found from Direct Numerical Simulations (DNS) that Eq. (3) with $C_S \sim 0.60$ is valid up to $Ri \sim 0.2$ at least. For $0 < Ri \lesssim 0.25$, $C_s'(Rf)$ decreases from 0.63 to 0.60, using the analytical expression (22) of Basu and Holtslag (2021) for $Pr(Ri)$. For Ri $\rightarrow 0$, we have Rf $\rightarrow 0$, then $\varepsilon_s' \rightarrow \varepsilon_S = 0.63 \sigma^2 S$. For Ri $\rightarrow 1$, Rf $\sim 0.25$, $\varepsilon_s' \rightarrow \varepsilon_N \sim 0.5 \sigma^2 N$. Therefore, Eq. (2) would





be an asymptotic expression valid for Ri of the order of 1 only. Eq. (3) removes an inconsistency in Eq. (2), since it wrongly indicates that $\varepsilon \to 0$ when $N \to 0$ for a given $\sigma^2$. If S = 0, i.e. if the source of the instability that generates turbulence is removed, then $\varepsilon = 0$ which makes more sense.

As discussed by Basu and Holtslag (2021, section 6.2) and Basu and Holtslag (2022, their Appendix 1), the derivation of Eq. (4) does not consider the fact that the steady state condition (also called "Full Equilibrium", Baumert and Peters, 2000) can

only be reached for a single value of Richardson number $Ri_s$, at least for large Reynolds numbers and large shear parameters $ST_L$ where $T_L$ is the inertial time scale defined as $TKE/\varepsilon$ (see, e.g., Mater and Venayagamoorthy, 2014). For $Ri < Ri_s$, TKE increases at subcritical $Ri$ and for $Ri > Ri_s$, TKE decreases (turbulence decays) at supercritical $Ri$ (e.g Baumert and Peters, 2000). From Large Eddy Simulation (LES) and DNS data, Schumann (1994) and Gerz et al. (1989) reported $Ri_S \approx 0.13$ for air, consistent with the value that can be deduced from Fig.1 of Mater and Venayagamoorthy (2014). Schumann (1994) re-

wrote the TKE budget equation as $dTKE/dt = (G - 1)(\varepsilon + B)$ where $G = P/(\varepsilon + B)$ is called the growth factor. $G = 1$ for FE conditions. By assuming, for simplicity, that $G$ depends only on $Ri$, Schumann proposed the empirical expression $G(Ri) = G_0^{(1-Ri/Ris)}$ with $G_0 = 1.47 \pm 0.13$ based on wind-tunnel data analysis. By using the same procedure as Basu and Holtslag (2021) from their equations (10) to (12) but starting with $= P/G - B$ , we get:

$$\varepsilon_s'' = \frac{0.63}{G^{1/2}}\left(1 - G\,R_f\right)^{1/2}\sigma^2 S \qquad\qquad (4')$$

Eq. (4') can also be directly obtained from Eq. (46a) into Eq. (10b) of Basu and Holtslag (2021). For $0 < Ri \lesssim 0.25$, $C_S'' = C_S/G^{1/2}\left(1 - G\,R_f\right)^{1/2}$ increases from 0.52 to 0.70, i.e. ~0.60 in average, for $Ri_S \approx 0.13$ and $G_0 = 1.47$. The Ri-dependence of $C_S''$ is thus only a small source of a dispersion for low $Ri$ values when comparing with other estimates.

From Baumert and Peters' (2000) results using a "Structural Equilibrium" approach (i.e. stationarity of ratios of turbulence characteristics) and based on laboratory and LES data (their figure 4), we can establish $\varepsilon_S = 0.15\sigma^2 S$ valid for $Ri \lesssim 0.25$.

This expression is obtained by combining $L_H/L_B = Ri^{1/2}$, $L_E/L_O = 4.2Ri^{3/4}$ and $L_E/L_B = 1.61\,Ri^{1/2}$ , where $L_E = \sqrt{\langle\theta'^2\rangle}/(d\theta/dz)$ and $L_O = \sqrt{\varepsilon/N^3}$ are the Ellison and Ozmidov scales, respectively. The constant differs very significantly (by a factor of 3 to 4 less) from the aforementioned estimates. If we use $L_E/L_O = 2.4\,Ri^{3/4}$ as proposed by Schumann (1994) for $Ri \leq 0.25$, we get $\varepsilon_S = 0.44\sigma^2 S$ with the same $L_E/L_B$ ratio. These expressions are more subject to experimental uncertainties and are thus not considered in this paper. In appendix (1), we propose an alternative derivation of Eq. (3)

suggesting $0.45 \leq C_S \leq 0.82$ . We retain the value of $C_S = 0.63$ for the comparisons between the models.

Following the spectral approach proposed by Weinstock (1981), Eq. (3) with $C_S$ equal to $C_N \approx 0.5$ can also be obtained from the integration of the 1D Kolmogorov (-5/3 slope) scalar kinetic energy spectrum over a spherical shell of radius $k_H$ instead of $k_B$ where $k_H$ ($k_B$) is the wavenumber corresponding to $L_H$ ($L_B$). For the context of radar measurements (e.g. Hocking et

al., 2016), Eq. (3) can also be obtained from the integration of the 1D transverse vertical velocity spectrum with a -5/3 slope for large (horizontal) wavenumbers ($k > k_H$) and a 0 slope for ($k < k_H$), both mathematical developments being equivalent. In essence, there is no contribution from an anisotropic buoyancy subrange.

Finally, $\varepsilon_S$ has the advantage that it can be evaluated entirely from the radar data, since the wind shear S can be estimated at the range and time resolutions of the radar, unlike $\varepsilon_N$ which requires $N^2$ to be obtained from in situ or Radio-Acoustic

Sounding System measurements.

### 3. The WPR-LQ-7 and methods of comparisons with UAV data

### 3.1 The WPR-LQ-7



The WPR-LQ-7 is a 1.3575 GHz Doppler radar. It has a phased array antenna composed of seven Luneberg lenses of 800 mm diameter. Its peak output power is 2.8 kW. It can be steered into five directions sequentially (i.e., after FFT operations), vertical and 14.2° off zenith toward North, East, South and West. The main radar parameters of the WPR-LQ-7 installed at Shigaraki MU Observatory since 2006 are given in Table 1.

| Parameter | |
|---|---|
| Beam directions | (0°,0°),(0°,14.2°), (90°,14.2°), (180°,14.2°), (270°,14.2°) |
| Radar frequency (MHz) | 1357 |
| Interpulse period ($\mu s$) | 80 |
| Subpulse duration ($\mu s$) | 0.67 |
| Pulse coding | 16-bit optimal complementary code |
| Range resolution (m) | 100 |
| Number of gates | 80 |
| Coherent integration number | 64 |
| Incoherent integration number | 18 |
| Number of FFT points | 128 |
| Acquisition time for one profile (s) (Antenna beam switched after FFT) | 59 s |
| Acquisition time of the mean profile (min) | 10 |
| Velocity aliasing ($ms^{-1}$) | 10.8 |

**Table 1: WPR-LQ-7 parameters in routine observation mode**

The acquisition time for one profile composed of 80 altitudes from 300 m AGL every 100 m in each direction is 59 sec after 18 incoherent integrations but for a total of 11.8 sec of observations for each direction (due to the intertwining between the directions). The time series are processed by automatic algorithms to remove outliers (e.g., bats, birds, airplanes) and ground clutter as far as possible. Low signals near and below the detection thresholds are removed, and profiles of atmospheric parameters (echo power, radial winds, half-power spectral width, horizontal and vertical winds) averaged over 10 min are made available for routine monitoring (http://www.rish.kyoto-u.ac.jp/radar-group/blr/shigaraki/data/). Because of the high data quality control, the 10-min averaged data are used to retrieve ε with the objective to assess the routine data for further analysis. The 1-min resolution data and those collected by the MU radar at a time resolution (sampling) of 24.57 s (~12.3 s) were used to help identify of atmospheric structures from height-time Signal to Noise Ratio (SNR) or echo power cross-sections, such as convective cells or Kelvin-Helmholtz billows. Table 2 shows the acquisition time, the range and transverse resolutions of the WPR-LQ-7 for the altitude range of comparisons and those of the MU radar for the data used in the present work.

### 3.2 The methods of comparison with DataHawk –derived $\varepsilon$

The DataHawk datasets were collected during two field campaigns, called the Shigaraki UAV Radar Experiments (ShUREX), in May–June 2016 and June 2017 at the Shigaraki MU observatory. The DataHawks were flying about 1-km away from the MU radar and the WPR-LQ-7. Kantha et al. (2017) described the instruments and configurations used during a previous ShUREX campaign in June 2015. The processing method used to retrieve ε from Pitot sensor data is not recalled here as it is described in detail by L18 for comparisons with MU radar data. The trajectories of the DataHawks being helicoidal upwards





or downwards, pseudo-vertical profiles of ε at a vertical sampling of ~5 m typically were obtained during the ascents and descents of the aircraft, from the ground up to a maximum altitude of ~4.5 km. Thirty-six DataHawk flights collected during the two campaigns provided ninety full or partial profiles used for the comparisons. Section 4 describes one of these flights with one full ascent (A1) and descent (D2) and one partial ascent (A2) and descent (D1). Three DataHawk flights collected during periods of precipitations contaminating the WPR-LQ-7 returns were rejected. The DataHawk-derived ε profiles were smoothed with a Gaussian window and resampled at the altitude of the radar gates to simulate the radar range resolution. The

degraded DataHawk 100-m resolution profiles are hereafter noted $\varepsilon_U$.

| | MU radar (during the campaigns) | WPR-LQ-7 (routine mode) |
|---|---|---|
| Acquisition time (s) (for one profile) | 24.57 every 12.3 s | 0.66 s (every 3.3 s) × NINCOH(18) = 11.8 s over 59 s |
| Range resolution (m) | 150 | 100 |
| Transverse resolution (m) (at z=2000 m) | ~100 | ~150 |

**Table 2: Time, range and transverse resolutions of the MU radar and WPR-LQ-7 for the dataset used in the present work. NINCOH refers to the number of incoherent integrations. The range resolution is $\Delta r = 1/2c\tau$ where c is the light speed and $\tau$ is the pulse duration and the transverse resolution is $2\theta_0 z$ where $\theta_0$ is half-power half width of the**

**effective (two-way) radar beam and z is altitude as defined in L18. The time series of MU radar signals are weighted by a Hanning window before FFT calculations.**

The WPR-LQ-7-derived $\varepsilon_{Lout}$, $\varepsilon_N$ and $\varepsilon_S$ profiles were computed at a time resolution of 10 to 30 minutes, i.e., by averaging up to 3 consecutive profiles that best correspond to each period of DataHawk ascent or descent. $\varepsilon_{Lout}$ was calculated with

$L_{out} = 70$ m in accordance with the best agreement from comparisons with the MU radar (L18) and the statistics of $L_{out}$ shown in section 5. The profiles of $N^2$ at a vertical resolution of 100 m were estimated from pressure and temperature profiles collected by the DataHawks. These profiles were used to obtain $\varepsilon_N$ (Eq. 2). $\varepsilon_S$ profiles were computed from Eq. (3) every 10 min from wind shear estimated from radar data and then averaged up to 30 min.

**3.3 Estimation of $\varepsilon$ from the Thorpe method applied to radiosonde data**

The Ozmidov scale $L_O = \sqrt{\varepsilon/N^3}$ is commonly assumed to be proportional to the Thorpe length $L_T \triangleq \langle d'^2 \rangle^{1/2}$, where $d'$ is the Thorpe displacement in the so-called Thorpe layer. Then, we have $L_O = cL_T$ and:

$$\varepsilon_T = c^2 L_T^2 N^3 \tag{5}$$

The literature is very divided on the value of c to use $(0.25 < c < 4)$ (see Kohma et al. 2019, for a review). Wijesekera and Dillon (1997) showed large temporal variations of $L_T/L_O$ from observations in the ocean. Large temporal variations were also

reported from DNS depending on the stage and source of turbulence (e.g., Fritts et al., 2016). An intermediate value of $c = 1$ is sometimes used by default (e.g. Kantha and Hocking, 2011). However, Mater et al. (2013) showed that $c \sim 1$ when the turbulent Froude number $Fr = \varepsilon/(N\,TKE)$ is near unity (at the transition between shear- and buoyancy dominated regimes). The basic $N^2$ for the Thorpe layers is generally estimated from the sorted potential temperature profile $(N_s^2)$ or from the r.m.s.





value of the fluctuations defined as the difference between the measured and sorted profiles ($N_{rms}^2$) (e.g. Smyth and Moum,
2001; Wilson et al., 2014).

Another scale, called the Corrsin scale is defined as $L_c = \sqrt{\varepsilon/S^3}$. It is the counterpart of the Ozmidov scale for shear flows
under neutral stratification conditions. Similarly, assuming $L_c = c'L_T$, we can write:

$$\varepsilon_{T'} = c'^2 L_T^2 S^3 \qquad (6)$$

Eq. (6) is thus a possible alternative to Eq. (5), when the Corrsin scale is smaller than the Ozmidov scale. These equations are
coherent with the results of Mater et al. (2013) who showed that $L_T$ scales with $(TKE)^{1/2}/S$ in the shear-dominated regime
and with $(TKE)^{1/2}/N$ in the buoyancy-dominated regime. Eq. (6) can also be justified and the parameter $c'$ can be estimated
as follows. The aforementioned ratios $L_E/L_O = 4.2\, Ri^{3/4}$ and $L_E/L_O = 2.4\, Ri^{3/4}$ found for weakly stratified flows ($Ri \leq$
0.25) by Baumert and Peters (2000) and Schumann (1994), respectively, may be representative of $L_T/L_O = 1/c(Ri)$ for flows
free of gravity wave motions. Indeed, $L_E = L_T$ is obtained for $d\theta/dz = cst$, which implies the absence of gravity waves. By
introducing the expressions of $L_T/L_O = 1/c(Ri)$ into eq. (5), we obtain Eq. (6) with $c'$ (hereafter noted $c'_{Sc}$) equal to $1/2.4 =$
0.41 or $c'$ (hereafter noted $c'_{BP}$) equal to $1/4.2 = 0.24$. Note that $c'$ is a constant while $c$ depends on $Ri$. For a shear-
dominated regime, from Figs. 3e, f, g, h of Mater and Venayagamoorthy (2014), we can deduce $0.25 < c' < 0.5$ typically
from DNS and $c' \sim 0.33$ from experimental data, which is very consistent with the other values. In Appendix 2, we show that
$c' = 0.28$ can be found from an alternative approach based on the inference of the turbulent Froude number from $L_E/L_O$ for
weakly stratified flow condition (Garanaik and Venayagamoorthy, 2019).

Eqs. (5) and (6) are equivalent if Eq. (5) is written as:

$$\varepsilon_T = c'^2 Ri^{-3/2} L_T^2 N^3 \qquad (7)$$

Eq.(3) and Eq. (6) have in common to be formally independent of $N^2$ when $Ri \lesssim 0.25$. If they were both confirmed by
experimental analysis, they would constitute a coherent whole.

## 4. Two case studies

Figure 1a shows the time-height cross-section of WPR-LQ-7 SNR (dB) at vertical incidence and a time and range resolution
of 1 min and 100 m, respectively on 18 June 2017 from 13:30 LT to 17:30 LT and in the altitude range [0.685-7.0 km] ASL
(ASL=AGL+0.385 km). Figure 1b shows the corresponding cross-section of MU radar echo power (dB) at vertical incidence
and a time resolution of ~12 sec after doing range imaging with Capon processing (e.g., Luce et al., 2017) in the altitude range
[1.275-7.0 km]. Radar echoes from a DataHawk, called "DH35" in reference to the flight numbering, are visible after ~14:30
and before ~15:40 LT on both images. They are the signatures of two ascents ('A1', 'A2') and two descents ('D1', 'D2') of a
DataHawk. Four red segments emphasize them in Fig. 1b. Incidentally, radar echoes from another DataHawk (DH36) can be
noted after 16:30 LT. A Vaisala RS92-SGP radiosonde, called "V6", was launched at 14:51 LT from the observatory. Its time-
height position is indicated by the blue line in Fig. 1b. It roughly coincides with A1.

An approximately 800 m deep enhanced echo power layer with S-shaped structures, signature of Kelvin-Helmholtz billows,
is clearly visible in both images in the altitude range [3.0-4.0 km] until ~16:00 LT at least. The layer is denoted by 'KHI' on
the images. DH35 crossed the layer four times during A1, D1, A2 and D2 between 15:00 and 15:30 LT. DH35 sampled the
most obvious case of K-H instability during the entire two campaigns. Since a necessary condition for the development of K-
H billows is Ri < Ric = 0.25 at the critical level, their observation suggests that it was fulfilled when sampled by the
instruments. Another focus will be given to a turbulent layer between 2100 and 2500 m, sampled twice by DH35 during A1
and D2, even though it is not clearly visible in the radar echo power images (Fig. 1). For this layer, Ri is expected to be $\gtrsim 1$
according to various estimates and layer properties described in Section 4.2.



The analysis of these two cases is made to illustrate the differences in the ε behavior of the three different radar models applied to two radars possibly at different Richardson numbers (Ri ≲ 0.25 and ≳ 1), compared to the DataHawk-derived ε.

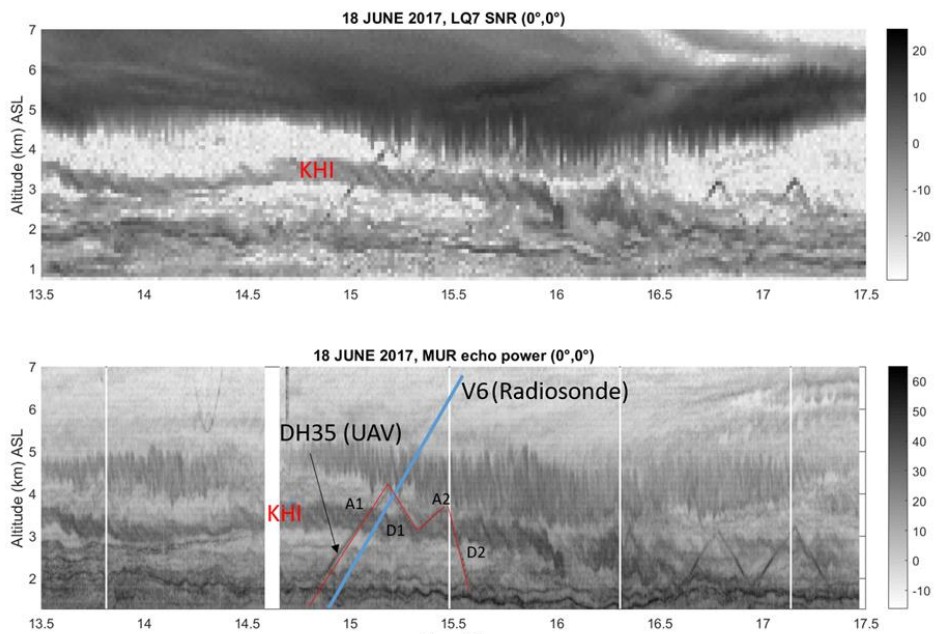


**Figure 1: (Top) Time-height cross-section of WPR-LQ-7 signal to noise ratio (dB) at vertical incidence on 18 June 2017 from 13:30 LT to 17:30 LT. (Bottom) The corresponding time-height cross-section of MU radar echo power (dB) in (high resolution) range imaging mode at vertical incidence. "A1", "D1", "A2" and "D2" refer to the consecutive ascents and descents of the DataHawk UAV (DH35) emphasized by the red lines. The blue line shows the time-altitude of the**

**radiosonde V6 launched at 14:51 LT from Shigaraki MU Observatory.**

**4.1 The K-H layer**

### 4.1.1 Comparisons between DataHawk-derived ε, $\varepsilon_{Lout}$, $\varepsilon_N$ and $\varepsilon_S$

Figure 2a shows the four DataHawk-derived ε profiles during A1, D1, A2 and D2 (dotted black lines) and the profiles of $\varepsilon_{Lout}$, $\varepsilon_N$ and $\varepsilon_S$ in the height range [2,000-3,900] m obtained from the WPR-LQ-7 data (red, blue, and green solid lines, respectively)

and MU radar data (red, blue, and green dashed lines, respectively). Figures 2c, 2d and 2e show the same information for the three models, but separately. For clarity and because they are very similar during D1, A2 and D2, the radar-derived ε profiles are shown for A1 (15:00-15:20 LT) only. Table 3a and the corresponding Figure 3 show ε, $\varepsilon_{Lout}$, $\varepsilon_N$ and $\varepsilon_S$ values averaged over the depth of the K-H layer for A1, D1, A2 and D2. The DataHawk-derived ε values peak in the range of the K-H layer and vary little during the ascents and descents over ~30 min: typically ~2 mWkg$^{-1}$. During A1, the DataHawk-derived ε

profile shows a narrower peak between 3200 and 3600 m. The DataHawk may have sampled a thinner region of the K-H layer (~400 m), perhaps associated with the edge of a K-H billow. This could also be the case for V6 as the Thorpe analysis suggests a ~300 m deep layer at the altitude of ~3.3 km (and an additional thinner layer around the altitude of ~3.6 km). If we exclude the difference in layer depth during A1, $\varepsilon_{Lout}$ and $\varepsilon_S$ estimated from both radars coincide very well with DataHawk-derived ε both in shape and levels during A1, D1, A2 and D2, with very similar variations in time (Table 3a and Fig. 3), indicating that

the two radar models are satisfactory and are equivalent in these circumstances. In contrast, the $\varepsilon_N$ profiles exhibit the worst agreement with DataHawk-derived ε near the center of the K-H layer where they show a minimum (Figs.2a and 2d, solid and dashed blue lines). This feature is similar to the one reported by L18 (their Figure 12) for a turbulent layer generated by a



convective instability at a mid-level cloud base. Table 3a and Figure 3 confirm that $\langle \varepsilon_N \rangle$ values are lower than the other estimates by a factor 2 to 3 approximately during A1, D1, A2, and D2 for both radars. This disagreement, occurring repeatedly

on the two radars, confirms the inadequacy of the $\varepsilon_N$ model for this layer.

### 4.1.2 Comparison between DataHawk-derived ε and $\varepsilon_T$

The altitude and depth of the turbulent layers identified by the Thorpe method from V6 and $\varepsilon_T$ (Eq. 5) with c=1 are shown by the dots and the solid vertical magenta segments, respectively, in Figs. 2a and 2f. In Fig. 2f, $\varepsilon'_T$ (Eq. 6) with $c' = 1$, $c'_{BP} = 0.24$ and $c'_{Sc} = 0.41$ are also shown for the K-H layer at 3.33 km and the turbulent layer at 2.37 km discussed in section 4.2. $N_s^2$

and $N_{rms}^2$ for the K-H layer are $7.7 \ 10^{-6} \ s^{-2}$ and $7.1 \ 10^{-6} \ s^{-2}$, respectively, i.e., $7.4 \ 10^{-6} \ s^{-2}$ in average. Because $L_T = 130 \ m$ in the K-H layer, we obtain $\varepsilon_T \approx 0.35 \ mWkg^{-1}$ which is about 7 times lower than DataHawk-derived ε (2.4 $mWkg^{-1}$) (Table 3a). The hypothesis that V6 passed through the K-H layer in a region where ε was much lower is not consistent with the low variability (stationarity) of the dissipation rates estimated from DataHawk and radar data for more than 40 minutes (see Table 3a and Fig. 3). We therefore assume that $\varepsilon_T$ must be $\approx 2.4 \ mWkg^{-1}$. To achieve this condition with

Eq. (5), we must have $c = 2.6$.

On the other hand, estimating $\varepsilon'_T$ (Eq. 6) requires to retrieve $S$ but there is no prescribed method to compute the vertical shear of horizontal wind from balloon data in the Thorpe layers. Here, we estimated $S$ from the difference of the wind vectors at the extremities of the Thorpe layer and from a linear interpolation of the zonal and meridional wind components in the Thorpe layer. We found $S =0.013 \ s^{-1}$ and $0.010 \ s^{-1}$, respectively, i.e., $0.0115 \ s^{-1}$ in average, so that $Ri \approx 0.055$. This value is close

to the mean value ($\langle Ri \rangle = 0.09$) obtained at a vertical resolution of 20 m (Fig. 2b). The relevance of $\varepsilon_T \approx 0.35 \ mWkg^{-1}$ obtained with $c = 1$ from Eq. (5) can be tested from Eq. (7) with $c = c'Ri^{-3/4} = 1$ using $c'_{BP} = 0.24$ and $c'_{Sc} = 0.41$. We get $Ri = 0.15$ and $Ri = 0.30$, respectively. These values are significantly larger than 0.055. For $S = 0.0115 \ s^{-1}$ and $c' = 1$, we get $\varepsilon_{T'} \approx 25.7 \ mWkg^{-1}$, i.e. about 11 times larger than DataHawk-derived ε. We must have $c' = 0.31$ to be consistent with the DataHawk-derived ε value. $c'_{Sc} = 0.41$ or $c'_{BP} = 0.24$ (and $c' = 0.28$ found in Appendix 2) reasonably meet the

necessary correction, giving credence to the validity of Eq. (6). As a corollary, for $Ri = 0.055$, we would get $c = c'Ri^{-3/4} = 2.7$ i.e. the required value of c for Eq. (5) to be valid. The various estimates of $\varepsilon_{T'}$ for the Thorpe layer are shown in Fig. 2f. Although based on a fragile hypothesis (ε from Thorpe analysis of the radiosonde data is equal to DataHawk-derived ε), Eq. (6) appears to be more adapted than the standard model (Eq. 5). It also has the major advantage over Eq.(5) that $c'$ is a true constant at least when $Ri < 0.25$, although its value remains to be defined more precisely.

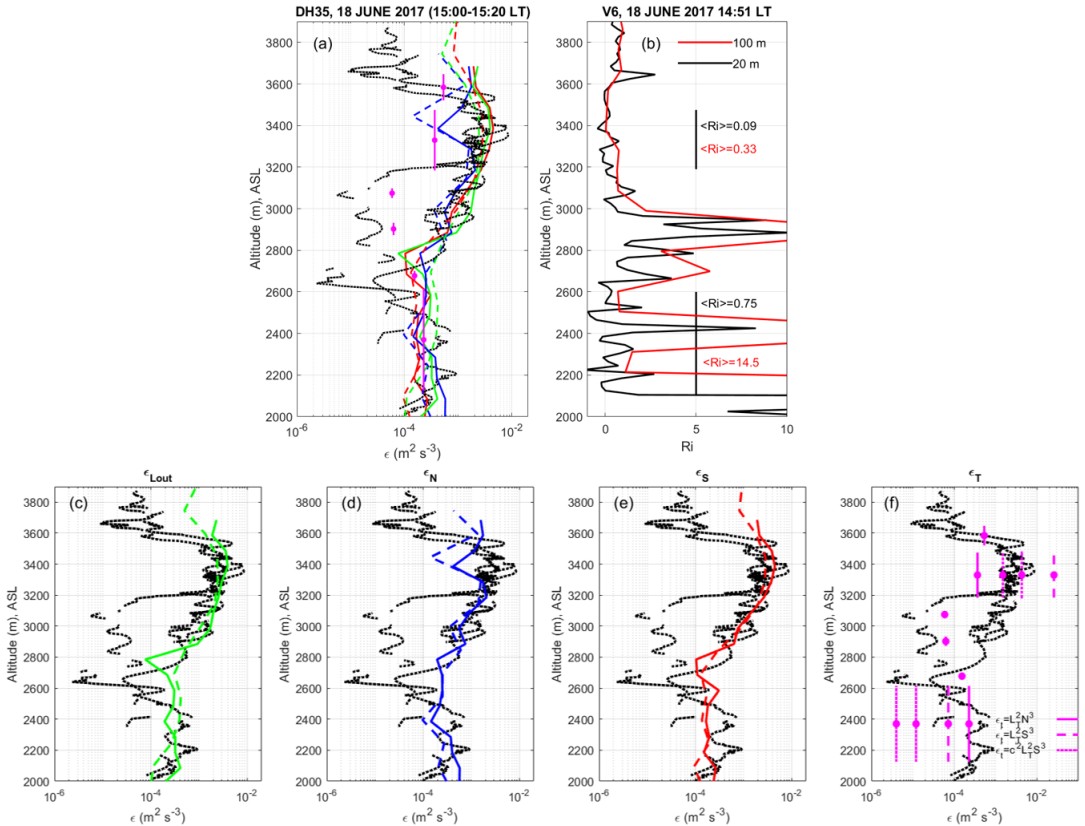

**Figure 2:** (a) DataHawk-derived $\varepsilon$ $(m^2 s^{-3})$ profiles in the height range [2000-3900] m during A1, D1, A2 and D2 of DH35 on 18 June 2017 (dotted black), $\varepsilon_S(LQ7)$ profile (solid red), $\varepsilon_{Lout}(LQ7)$ profile (solid green), $\varepsilon_N(LQ7)$ profile (solid blue), $\varepsilon_S(MU)$ profile (dashed red), $\varepsilon_{Lout}(MU)$ profile (dashed green), $\varepsilon_S(MU)$ profile (dashed blue) derived from radar data between 15:00 and 15:20 LT. Magenta dots and lines show $\varepsilon_T$ (Eq. 5) with c=1, the depth and altitude of the Thorpe layers. (b) Richardson number profiles estimated from RS92-SGP Vaisala radiosonde V6 data at a vertical resolution of 20 m (black) and 100 m (red). (c,d,e,f) Same as (a) but with separate plots for each model ($\varepsilon_{Lout}, \varepsilon_N, \varepsilon_S, \varepsilon_T$), respectively. (f) shows the results in magenta for $\varepsilon_T$ (Eq. 5) with c=1 (solid line), $\varepsilon'_T$ (Eq. 6) with $c' = 1$ (dashed line) and $\varepsilon'_T$ (Eq. 6) with $c' = 0.41$ and $c' = 0.24$ (dotted line).



| K-H | $< \varepsilon_U >$ | $< \varepsilon_{Lout} >$ | $< \varepsilon_S >$ | $< \varepsilon_N >$ | $< \varepsilon_{Lout} >$ | $< \varepsilon_S >$ | $< \varepsilon_N >$ | $\varepsilon_T$ |
|---|---|---|---|---|---|---|---|---|
| | | MU radar | MU radar | MU radar | LQ-7 | LQ-7 | LQ-7 | (c=1) |
| A1 | 2.42 | 2.43 | 2.38 | 0.94 | 2.62 | 2.81 | 1.30 | 0.37/0.32* |
| D1 | 1.91 | 2.11 | 2.30 | 0.83 | 2.57 | 2.22 | 1.30 | |
| A2 | 2.54 | 2.06 | 2.53 | 0.81 | 2.52 | 2.62 | 0.61 | |
| D2 | 3.14 | 6.56 | 6.04 | 1.75 | 6 .51 | 4.84 | 1.48 | |

| TL | $< \varepsilon_U >$ | $< \varepsilon_{Lout} >$ | $< \varepsilon_S >$ | $< \varepsilon_N >$ | $< \varepsilon_{Lout} >$ | $< \varepsilon_S >$ | $< \varepsilon_N >$ | $\varepsilon_T$ |
|---|---|---|---|---|---|---|---|---|
| | | MU radar | MU radar | MU radar | LQ-7 | LQ-7 | LQ-7 | (c=1) |
| A1 | 0.39 | 0.41 | *0.09** | 0.18 | 0.37 | 0.24 | 0.33 | 0.19/0.23** |
| D2 | 0.12 | 0.10 | 0.05 | 0.08 | 0.11 | 0.09 | 0.12 | |

*: this low value is due a MU radar-derived wind shear about twice smaller than LQ7 and Balloon-derived wind shear. Its is doubtful and will affect $L_H$ and $L_c$ in Table 4 **:*(sorted/r.m.s.)*

**Table 3: Mean values of TKE dissipation rates ($mW\ kg^{-1}$) according to the different models and instruments for the**

**K-H layer (top) and for the turbulent layer (TL) between 2100 and 2500 m (bottom).4.1.3 Comparison with $\varepsilon_{T'}$**

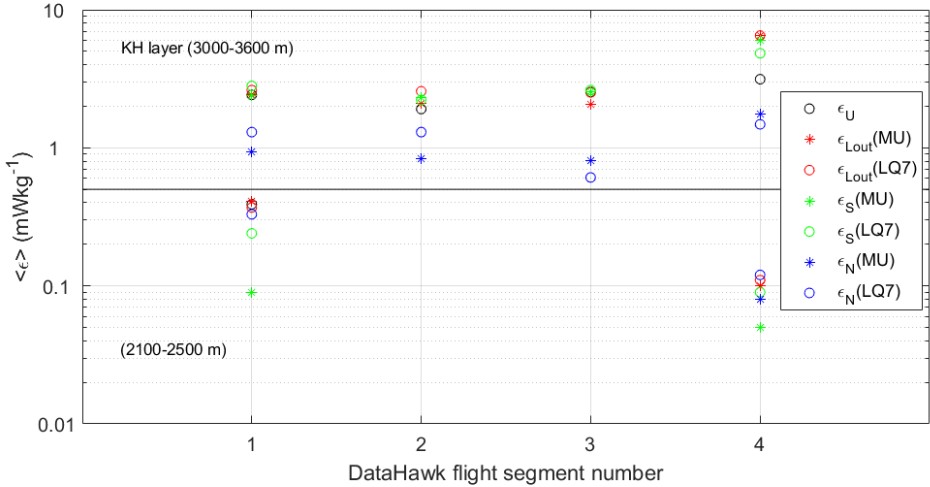

**Figure 3: Graphical representation of the averaged estimates of TKE dissipation rates ($mWkg^{-1}$) ($1\ mWkg^{-1} = 10^{-3}m^2s^{-3}$) shown in Table 3 for the K-H layer (3000-3600 m) sampled 4 times (top) and the layer between 2100 m and 2500 m sampled twice during A1 (segment n° 1) and D2 (segment n°4) (bottom). The horizontal black line at 0.5**

**$mWkg^{-1}$ separates the two cases for clarity.**

**4.1.3 Comparison of turbulence scales estimated from radar data.**

Table 4a shows the Hunt, Corrsin, Buoyancy, and Ozmidov scales for the K-H layer calculated from WPR-LQ-7 and MU radar-derived $\varepsilon$, $\sigma$ and $S$ during A1, D1, A2, and D2. $N^2$ is computed from balloon data at the radar resolutions (100 m for the

WPR-LQ-7 and 150 m for the MU radar). Once the scales are calculated, they are averaged over the altitude range 3000-3600 m of the K-H layer to compare them with the Thorpe length. All the radar-derived scales reveal the same behaviors between the segments A1, D1, A2, and D2 and do not show substantial differences between the radars, reinforcing the reliability of numerical results and their interpretations. Only the average values on A1, D1, A2 and D2 are discussed. We get $\langle L_H \rangle$=46 m,



$\langle L_C \rangle$ = 38 m, $\langle L_B \rangle$=105 m and $\langle L_O \rangle$ = 131 m. $\langle L_H \rangle$ and $\langle L_C \rangle$ are substantially smaller than $\langle L_B \rangle$ or $\langle L_O \rangle$ indicating the latter

should not be the scales to consider, as expected from the analysis of section 4.1.1. Depending on the flight segment (A1, D1,

A2, D2), $0.65 < \langle L_O \rangle / L_T = c < 1.56$ and ~1 in average. In contrast, we get $\langle L_C \rangle / L_T = c' \approx 0.23 - 0.37$. It is smaller than

$c'_{Sc}$ = 0.41 but close to $c'_{BP}$ =0.24 and the value obtained in Appendix 2 ($c'$ = 0.28) and the needed value 0.31. We obtain

$((\langle L_H \rangle / \langle L_B \rangle))^2 = \langle Ri \rangle = 0.19$ and $(c'/c)^{4/3} = \langle Ri \rangle = 0.28$. Both radar estimates are significantly larger than $Ri$ estimated

from balloon data with the Thorpe analysis but are close to $\langle Ri \rangle$=0.33 estimated from balloon data at the vertical resolution of

100 m (Fig. 2b). The quantitative disagreements result mainly from comparisons between estimates made with different

techniques (radar and in situ) and resolutions. Nevertheless, it is interesting to note that these comparisons tend to corroborate

the conclusions obtained from in-situ measurements alone (section 4.1.2).

**4.2 The turbulent layer between 2100 and 2500 m**

Using the same methods as for the K-H layer, we obtain $\langle Ri \rangle$ = 0.75 (14.5) at a vertical resolution of 20 (100) m from V6

data (Fig. 2b), Ri ≈ 2.0 from Thorpe analysis of V6 in the altitude range [2100-2500 m] (not shown) and $\langle Ri \rangle$ = 4.6 from $N^2$

calculated at a vertical resolution of 100 m from V6 data and S calculated from WPR-LQ-7 data during A1 and D2 (Table 4b).

Therefore, the Richardson number strongly varies according to the method and data used but all the estimates are consistent

with a Ri value significantly larger than for the K-H layer and likely larger than 1 (Section 4.1). Therefore, the weakly stratified

condition ($Ri < 0.25$) for which the alternative Eq. (3) and Eq. (6) are valid is likely not verified for this layer. DataHawk-

derived ε is about one order of magnitude lower than for the K-H layer: ~0.2 mWkg$^{-1}$ in average (Table 3b). The mean value

is less reliable during D2. Many values of DataHawk-derived ε are missing because the algorithm did not detect a -5/3 subrange

in the velocity spectra (see L18). Figure 2a, Table 3b and Figure 3 show that $\varepsilon_{Lout}$ and $\varepsilon_N$ and their mean values derived from

both radars and $\varepsilon_T$ are close to each other (within a factor less than ~2) and are very consistent with DataHawk-derived ε. All

the radar and DataHawk estimates show together a temporal decrease by a factor 3 to 4 in the 40 minutes between A1 and D2

(Fig. 3) giving credence that the agreements between the various estimates during A1 and D2 are not fortuitous. The temporal

decrease of ε is consistent with a decaying turbulence when $Ri > 1$. $\langle \varepsilon_S \rangle$ shows the largest discrepancies with DataHawk-

derived ε perhaps because the model is not valid for large $Ri$ values. $\langle L_H \rangle$ and $\langle L_C \rangle$ (136 m and 202 m, respectively) exceeds

$\langle L_B \rangle$ = 60 m and $\langle L_O \rangle$ = 56 m which are close to $L_T$(64 m) (Table 4b). Therefore, $\langle L_H \rangle$ and $\langle L_C \rangle$ should not be the turbulence

scales to consider. From the Thorpe analysis, $N^2 \approx 1.4 \ 10^{-5} \ s^{-2}$ and $S \approx 0.0026 \ s^{-1}$ ($Ri \approx 2$). From eq. (5) with $c = 1$,

$\varepsilon_T \approx 0.2 \ mWkg^{-1}$ (Table 3b), i.e. very close to the mean value of DataHawk-derived ε or only twice lower than the value

during A1. It is consistent with the fact that $L_T$ can be assimilated to $L_O$. From Eq.(6) with $c'_{Sc}$ = 0.41, $c'_{BP}$ =0.24 and $Ri = 2$,

we obtain $\varepsilon'_T \approx$ 0.012 and 0.004 $mWkg^{-1} \ll 0.2 mWkg^{-1}$ . The various estimates of $\varepsilon'_T$ are shown in Fig. 2f. As expected $\varepsilon'_T$

fails because it is expected to be valid for $Ri < 0.25$ only.

**5. Statistical analysis**

**5.1 Justification of $L_{out}$=70 m**

Figure 4 shows the histogram of $\log_{10}(L)$ where L = $\langle \sigma^2 \rangle^{3/2} / \varepsilon_U$ for $\langle \sigma^2 \rangle^{3/2} > 0.01$ as in L18 obtained from the WPR-LQ-7

from data collected during 36 flights (corresponding to 90 profiles). The peak of the distribution has a mean (median) value

of 67 m (71 m). These values are almost identical to those obtained from comparisons with MU radar, i.e. 75 m (61 m) (Fig.

7a of L18). This result seems to indicate that the empirical expression $\varepsilon_{Lout}$ with $L_{out}$=70 m is not specific to the MU radar,

but at least to any radar with similar resolution volume (Table 2). However, the acquisition time of the MU radar and WPR-

LQ-7 data differs by a factor ~2.5 (Table 2). It is likely fortunate that the statistical values of $\varepsilon_{Lout}$ (and thus $\sigma$) coincide so

well.





| KH | $< L_H >$ LQ-7/MU | $< L_C >$ LQ-7/MU | $< L_B >$ LQ-7/MU | $< L_O >$ LQ-7/MU | $L_T$ |
|---|---|---|---|---|---|
| A1 | 42/46 | 32/37 | 70/90 | 70/103 | |
| | 44 | 34 | 80 | 87 | 130 |
| D1 | 52/41 | 45/31 | 69/90 | 68/101 | |
| A2 | 43/36 | 34/26 | 144/89 | 206/100 | |
| D2 | 60/49 | 56/40 | 154/130 | 228/179 | |
| Mean | 46 | 38 | 105 | 131 | |

(a) $(< L_H >/< L_B >)^2 =< Ri >= 0.19$ during A1

| TL | $< L_H >$ LQ-7/MU | $< L_C >$ LQ-7/MU | $< L_B >$ LQ-7/MU | $< L_O >$ LQ-7/MU | $L_T$ |
|---|---|---|---|---|---|
| A1 | 69*/204 | 72*/331 | 39/80 | 31/82 | |
| | 136 | 202 | 60 | 56 | 64 |
| D2 | 55/91 | 49/104 | 32/43 | 22/34 | |
| Mean | 105 | 139 | 49 | 42 | |

(b) ∗: doubtful (see table 3)  $(< L_H >/< L_B >)^2 =< Ri >= 4.6$ during A1

**Table 4: Mean values of Hunt, Corrsin, Buoyancy and Ozmidov scales for the K-H layer (a) and for the turbulent layer (TL) between 2100 and 2500 m (b).**

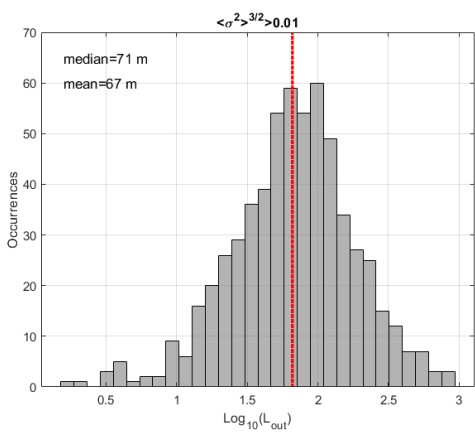


**Figure 4: Histogram of $log_{10}(L_{out})$ for $\langle \sigma^2 \rangle^{3/2} > 0.01$ as in L18 for MU radar data.**

### 5.2 Statistical evaluation of the models from comparisons with $\varepsilon_U$

Figures 5a, b, c show the scatter plots of $\log_{10}(\varepsilon_U)$ vs $\log_{10}(\varepsilon_{Lout})$, $\log_{10}(\varepsilon_S)$, and $\log_{10}(\varepsilon_N)$ with $L_{out} = 70$ m, the shear estimated from WPR-LQ-7 data and N from DataHawk data at a height resolution of 100 m. Figures 5d and 5e show the

results assuming a constant shear ($\langle S \rangle = 7.7$ ms$^{-1}$km$^{-1}$) and a constant N ($\langle N^2 \rangle = 6.7$ $10^{-5}$ s$^{-2}$). Of course, Figs. 5d and 5e differ only in the constant 0.64/0.5.





The correlation coefficients are fortuitously ~0.66 for all the cases except for $\varepsilon_N$ for which the correlation is ~0.60 only. This is an additional clue of the inadequacy of $\varepsilon_N$. The red lines show the results of linear regressions after rejecting dissipation rate values smaller than $1.6 \times 10^{-5}$ m$^2$s$^{-3}$ as in L18, even though the quantitative threshold has no reason to be the same since

the comparison methods differ. The slope of the regression line between $\log_{10}(\varepsilon_{Lout})$ and $\log_{10}(\varepsilon_U)$ is ~1.0 (Fig. 5a) confirming the statistical $\sigma^3$ dependence of $\varepsilon$ when no discrimination is made on the conditions under which turbulence occurs. The regression slope obtained with $\log_{10}(\varepsilon_N)$ or $\log_{10}(\varepsilon_S)$ for S = cst or N = cst (Fig. 5d, 5e) is 0.60, i.e., close to 2/3, as expected because the two models vary as $\sigma^2$. However, the regression slope between $\log_{10}(\varepsilon_U)$ and $\log_{10}(\varepsilon_N)$ with measured N (Fig. 5c) is significantly lower than 0.66 (0.50) and the regression slope between $\log_{10}(\varepsilon_U)$ and $\log_{10}(\varepsilon_S)$ with measured S

(Fig. 5b) is significantly larger than 0.66 (0.73). The regression slope between $\log_{10}(\varepsilon_U)$ and $\log_{10}(\varepsilon_N)$ using MU radar data was 0.55 (L18), i.e. virtually identical to the present case (Fig. 5c). All the regression slopes depend on the quantitative threshold on $\varepsilon$ and, in Fig. 5a, it varies from ~0.9 to ~1.1 for different thresholds excluding small values. However, all other slope estimates vary in concert so that the observed trends remain valid for a different threshold.

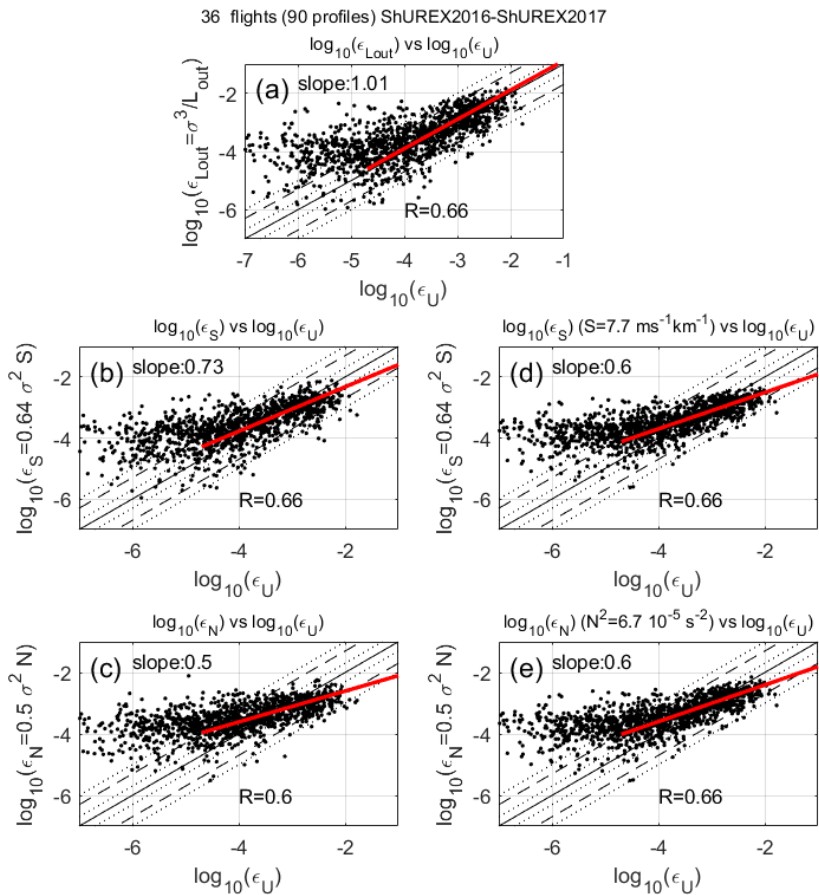

**Figure 5: Scatter plots of $log_{10}(\varepsilon_U)$ vs (a) $log_{10}(\varepsilon_{Lout})$, (b) $log_{10}(\varepsilon_S)$, (c) $log_{10}(\varepsilon_N)$, (d) $log_{10}(\varepsilon_S)$ with $S = cst$, (e) $log_{10}(\varepsilon_N)$ with $N = cst$. The red lines are the result of a line regression (whose slope value is indicated in the insert) for $\varepsilon_U > 1.6\ 10^{-5}\ m^2 s^{-3}$ and R is the correlation coefficient.**





(1) A regression slope between $\log_{10}(\varepsilon_U)$ and $\log_{10}(\varepsilon_S)$ that is closer to 1 than the slope between $\log_{10}(\varepsilon_U)$ and $\log_{10}(\varepsilon_N)$

indicates that $\varepsilon_S$ provides estimates more consistent with $\varepsilon_{Lout}$ than $\varepsilon_N$. Figures 6a and 6b show a comparison between the

radar models, i.e., $\log_{10}(\varepsilon_{Lout})$ vs $\log_{10}(\varepsilon_N)$ and $\log_{10}(\varepsilon_{Lout})$ vs $\log_{10}(\varepsilon_S)$, respectively, for $\varepsilon_U > 1.6\ 10^{-5}$ m$^2$s$^{-3}$. The red

lines are the results for N = cst and S = cst. The blue lines show the results of a linear regression. The blue and red lines

obviously coincide (slope=0.60) for $\varepsilon_N$. A slope of 0.77 is obtained with $\varepsilon_S$ indicating a greater equivalence between $\varepsilon_{Lout}$ and

$\varepsilon_S$, as expected from Fig. 5b. Consequently, our results suggest that $\varepsilon_S$ is more relevant than $\varepsilon_N$ and should be used instead of

$\varepsilon_N$ for operational use if the empirical model $\varepsilon_{Lout}$ is not chosen.

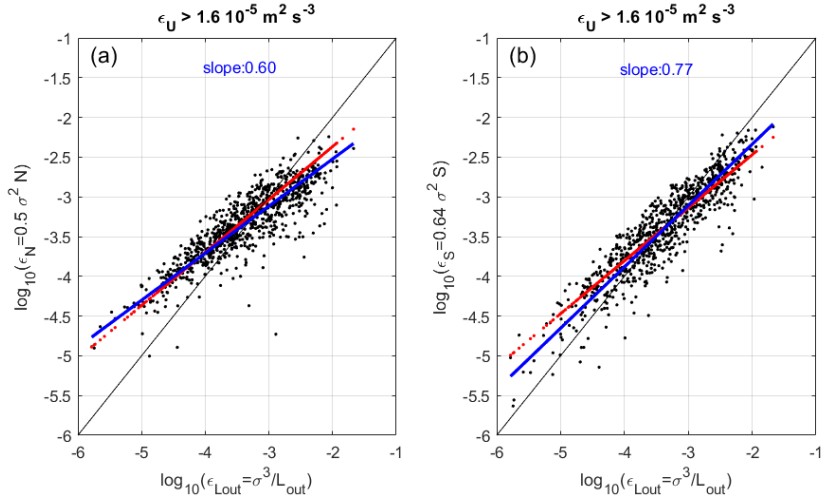

**Figure 6: Scatter plots of (a) $log_{10}(\varepsilon_{Lout})$ vs $log_{10}(\varepsilon_N)$ and (b) $log_{10}(\varepsilon_{Lout})$ vs $log_{10}(\varepsilon_S)$ for $\varepsilon_U > 1.6\ 10^{-5}\ m^2 s^{-3}$.**
**The red lines show the results for $N = cst$ (a) and $S = cst$ (b). The blue lines show the results of a linear regression**
**(whose slope value is indicated in the insert).**

(2) We checked that generating a normal random distribution of N or S with a mean and standard deviation similar to the

observed distributions produced a regression slope close to 2/3, similar to Fig. 5d and 5e. Therefore, the observed slopes with

measured S and N (Fig. 5b, 5c) must reveal a statistical dependence of σ with 1/N and S, respectively. The equivalence

between $\varepsilon_{Lout}$ and $\varepsilon_S$ described in Section 4 for the K-H layer implies that σ is simply proportional to S (σ = 0.64 $L_{out}$S) if

$L_{out}$=cst. There is a canonical value of $L_{out}$ (~70 m), but since $L_{out}$ is not constant and is unknown (and can vary by two

orders of magnitude at least, Fig. 4), the correlation between σ and S can only be established for a fixed value of $L_{out}$ (and for

any other variable on which σ depends). Figure 7 shows the same information as Fig.6 but after dividing by $σ^2$ to remove the

self-correlation between the variables and to show the relationship between $\log_{10}(σ/L_{out})$ and $\log_{10}(0.5\ N)$ (Fig. 7a) and

between $\log_{10}(σ/L_{out})$ and $\log_{10}(0.64\ S)$ (Fig. 7b). The two scatter plots show negative and positive correlation coefficients

(-0.36 and 0.12, respectively). The correlations are weak but significant according to the P test. If no threshold on $\varepsilon_U$ is applied,

the correlation coefficients are -0.26 and +0.22. This suggests that σ increases to some extent as S increases and N decreases.

It is quite intuitive, but, to our knowledge, this is the first study to suggest and highlight this. The results may reveal a

Richardson number dependence. Figure 8a shows scatter plots of σ vs $Ri_{100}^{1/2}$ where $Ri_{100}$ ($S_{100}$) now explicitly refers to the

Richardson number (shear) calculated at the vertical resolution of 100 m. The red (black) dots show the results without and

with a threshold on $S_{100}$ ($S_{100} > 5$ ms$^{-1}$km$^{-1}$), respectively. The (negative) correlation coefficient is slightly stronger with

the threshold (-0.34 instead of -0.21). The high values of $Ri_{100}$ are mainly associated with a weak shear ($S_{100} <$

$5$ ms$^{-1}$km$^{-1}$) and with the largest variability in σ (Fig 8.a). However, this property may not be significant because the





uncertainty on Ri increases as the shear tends to zero. Therefore, we focus on the scatter plot obtained with the threshold on

the shear (black dots). It seems to show a linear dependence between $\log_{10}(\sigma)$ and $\log_{10}(Ri_{100}^{-1/2})$, at least down to $\log_{10}(Ri_{100}^{1/2})$ $\approx -0.2$ i.e. for $Ri_{100} < 0.4$. Attempts of linear regression analysis do not confirm the linear trend, likely due to the strong dispersion and weak correlation. However, the time series obtained from the concatenation of all the profiles of $\log_{10}(Ri_{100}^{-1/2})$ and $\log_{10}(\sigma)$ after removing their mean values reveal a more obvious dependence between the two variables (Fig. 8b). The curves reveal similar variations and dynamics, especially for records [0-200], compatible with $\sigma^2$ inversely proportional to

$Ri_{100}$, at least to a first approximation. For $\log_{10}(Ri_{100}^{1/2}) \lesssim -0.2$, i.e. for low values of $Ri_{100}(< 0.4)$, $\log_{10}(\sigma)$ appears to have very little dependence with $\log_{10}(Ri_{100}^{1/2})$. If meaningful, it would be consistent with the fact that N does not play a significant role for low Ri values, as suggested by the $\varepsilon_S$ model.

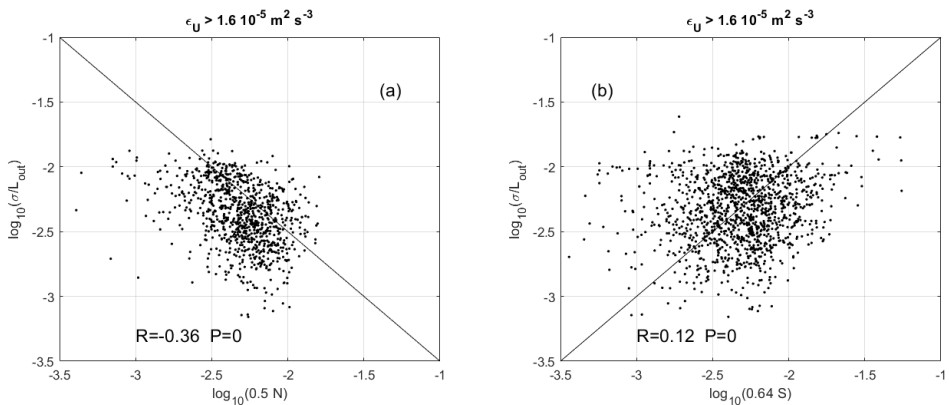

**Figure 7: Scatter plots of (a) $log_{10}(0.5\,N)$ vs $log_{10}(\sigma/Lout)$ and (b) $log_{10}(0.64\,S)$ vs $log_{10}(\sigma/Lout)$ for $\varepsilon_U >$ $1.6\,10^{-5}\,m^2s^{-3}$. R is the correlation coefficient and P is the result of the P test.**

Figure 9 shows the scatter plots of $\log_{10}(\varepsilon_{Lout}/\varepsilon_U)$, $\log_{10}(\varepsilon_N/\varepsilon_U)$, $\log_{10}(\varepsilon_S/\varepsilon_U)$ vs $\log_{10}(Ri_{100})$ applying two thresholds on $\varepsilon_U$: $1.6\,10^{-5}\,m^2s^{-3}$ in Fig. 9a, b, c and $5.0\,10^{-4}\,m^2s^{-3}$ in Fig. 9d, e, f. The latter is introduced to analyze the dependence of

the results on the levels of considered dissipation rates. The red and blue curves show the values averaged in bandwidths of 0.3 from $\log_{10}(Ri_{100}) = -1.7$. For $\varepsilon_U > 1.6\,10^{-5}\,m^2s^{-3}$, the mean curves of $\log_{10}(\varepsilon_{Lout}/\varepsilon_U)$ and $\log_{10}(\varepsilon_S/\varepsilon_U)$ does not reveal a significant dependence with $\log_{10}(Ri_{100})$, at least up to $\log_{10}(Ri_{100}) \sim 1$, and are almost identical and close to 0 (Fig. 9a, c). Therefore, the applicability of the two models does not seem to depend significantly on the Richardson number on average. For $\varepsilon_U > 5.0\,10^{-4}\,m^2s^{-3}$, the curves produced by two models remain close and almost unchanged for

$\log_{10}(Ri_{100}) < 0$ (Fig. 9d, f). However, the mean values of $\log_{10}(\varepsilon_S/\varepsilon_U)$ now tend to decreases as $\log_{10}(Ri_{100})$ increases. Therefore, when $\log_{10}(Ri_{100}) > 0$, $\varepsilon_S$ tends to underestimate $\varepsilon_U$ when $\varepsilon_U$ exceeds $\sim 5.0\,10^{-4}\,m^2s^{-3}$ and inversely when $\varepsilon_U < 5.0\,10^{-4}\,m^2s^{-3}$. The fact that we experimentally observe that $\varepsilon_S$ is not appropriate for large values of $Ri_{100}$ is consistent with the expected domain of applicability of the model even if it is not clear why it is in this way. For $\log_{10}(Ri_{100})<0$, $\log_{10}(\varepsilon_N/\varepsilon_U)$ is less than 0 and decreases as $\log_{10}(Ri_{100})$ decreases for both thresholds (Fig. 9b, e). This experimental

observation is a confirmation of the inadequacy of $\varepsilon_N$ when the Richardson number is low. The results with $\varepsilon_{Lout}$ are difficult to interpret when $\log_{10}(Ri_{100}) > 0$. The model is consistent with $\varepsilon_S$ when $\varepsilon_U > 1.6\,10^{-5}\,m^2s^{-3}$ (Fig. 9c) and seems to be more consistent with $\varepsilon_N$ than with $\varepsilon_S$ when $\varepsilon_U > 5.0\,10^{-4}\,m^2s^{-3}$ (Fig. 9e, f). It may be vain to interpret the properties of this model, since it is only an empirical model for which $L_{out} = 70$ m represents only a canonical value of a function of multiple variables including the shear and N.


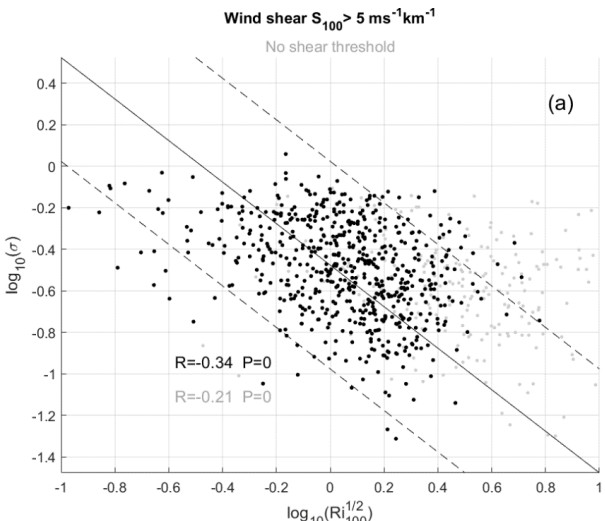

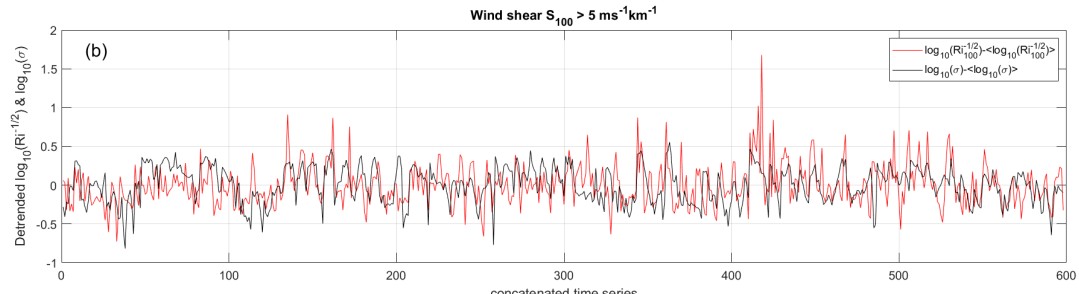

Figure 8: (a) Scatter plots of $log_{10}\left(Ri_{100}^{1/2}\right)$ vs $log_{10}(\sigma)$ for $\varepsilon_U > 1.6\ 10^{-5}\ m^2s^{-3}$ without threshold on shear (grey) and for $S_{100} > 5ms^{-1}km^{-1}$. (b) The corresponding time series of $log_{10}\left(Ri_{100}^{-1/2}\right)$ (grey) and $log_{10}(\sigma)$ (black) after subtracting their mean.

## 6. Conclusions

The objective of this work was to test the suitability of TKE dissipation rate models based on Doppler radar spectral width measurements from comparisons with in-situ estimates ($\varepsilon_U$) derived from high-resolution Pitot tube measurements aboard DataHawk UAVs. We showed that:

(1) the models applied to the 46.5 MHz MU VHF radar by L18 produce statistically identical results on the 1.357 GHz WPR-LQ-7:

(a) The empirical model $\varepsilon_{Lout} = \sigma^3/L_{out}$ with $L_{out}$ ~70 m (as for the MU radar) provides the best statistical agreement with $\varepsilon_U$ at least for $\varepsilon \gtrsim 2\ 10^{-5}\ m^2s^{-3}$ (Table 2). If $L_{out}$ really depends on the size of energy containing eddies, it is then independent of radar parameters (assuming $\sigma^2$ is true indication of $\langle w^2 \rangle$ in both radars).

(b) The model $\varepsilon_N$ predicting a $\sigma^2N$ law for stably stratified conditions fails to reproduce $\varepsilon_U$. The biases are nearly quantitatively identical to those obtained with the MU radar: $\varepsilon_N$ tends to overestimate when $\varepsilon_U < $ ~5 $10^{-4}\ m^2s^{-3}$ and to underestimate when $\varepsilon_U > $ ~5 $10^{-4}\ m^2s^{-3}$.




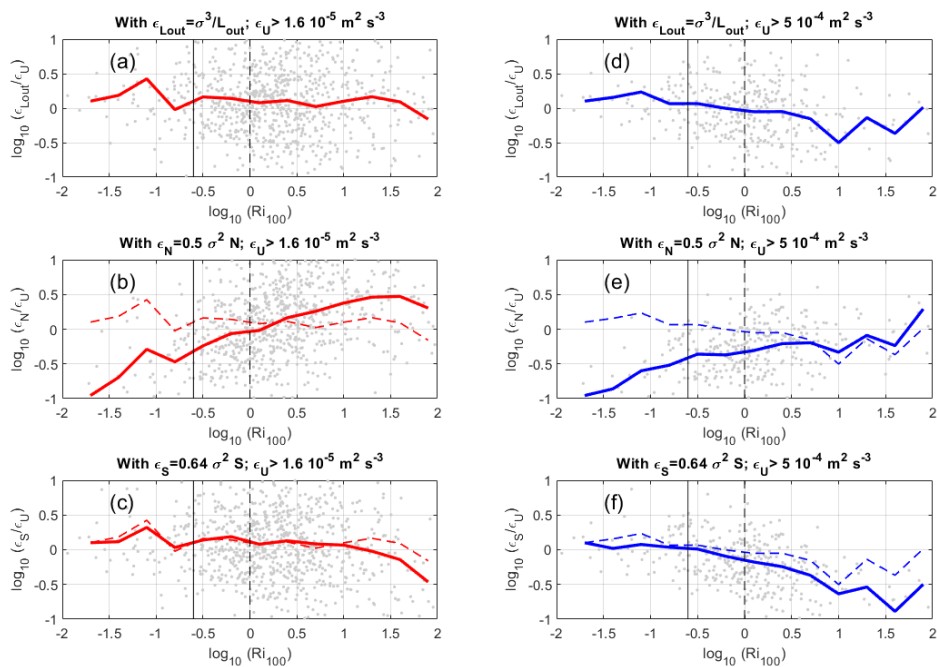

**Figure 9: Scatter plots of** $log_{10}(Ri_{100})$ **vs (a),(d)** $log_{10}(\varepsilon_{Lout}/\varepsilon_U)$, **(b),(e)** $log_{10}(\varepsilon_N/\varepsilon_U)$, **(c), (f)** $log_{10}(\varepsilon_S/\varepsilon_U)$ **for** $\varepsilon_U >$
$1.6\,10^{-5}\,m^2 s^{-3}$ **and** $\varepsilon_U > 5.0\,10^{-4}\,m^2 s^{-3}$, **respectively.**


(2) Applying $\varepsilon_N$ to both radars to a turbulent layer attributed to a K-H instability with Ri <0.25 strongly underestimates $\varepsilon_U$ in the core of the layer when $N^2$ is minimum. On the other hand, in agreement with the statistical results, $\varepsilon_N$ provided values consistent with the other estimates in a turbulent layer likely associated with larger Ri ($\gtrsim$ 1). These two observations are rather consistent with the domain of validity of $\varepsilon_N$ according to the theoretical derivations (Eq. 4) leading to the newly

introduced expression of $\varepsilon_S$ expected to be valid for weakly stratified or strongly sheared conditions (e.g., Basu et al., 2021).

(3) The application of $\varepsilon_S$ to the K-H layer (Ri <0.25) using both radars leads to a good agreement with $\varepsilon_U$. Its application to the turbulent layer associated with larger Ri slightly underestimates $\varepsilon_U$, again in accordance with Eq. (4).

(4) The statistical comparisons between $\varepsilon_S$ and $\varepsilon_U$ using all data show much better agreement than between $\varepsilon_N$ and $\varepsilon_U$, although a bias of the same nature as that observed with $\varepsilon_N$ is also noted, but to a lesser degree. Empirical $\varepsilon_{Lout}$ remains the

most consistent model compared with $\varepsilon_U$. $L_{out}$ ~70 m is likely a canonical value that results from all the hidden contributions of the various parameters that a most general (and unknown) model should include.

(5) The equivalence between $\varepsilon_S$ and $\varepsilon_U$ for the K-H layer associated with a low Ri necessarily implies that $\sigma$ is proportional to S: $\sigma$~0.64 $L_{out}$S. For all the layers with the same value of $L_{out}$, then $\sigma$ linearly depends on S. This is a necessary condition if agreement is observed with two models that predict a $\sigma^3$ and a $\sigma^2$ dependence. For a wide distribution of $log_{10}(L_{out})$ as in

Fig. 4, that includes values for all Ri, this linear dependence should be strongly "blurred" because $L_{out}$ is variable and Ri is not necessarily low. Moreover, an additional source of dispersion is that the wind shear calculated at the radar resolution $S_{100}$ and at a time resolution of 10-30 min is not necessarily the most effective shear to be considered, because S is a scale-dependent parameter (in the same way as Ri). As a result, a very weak, but yet significant, correlation between $\sigma$ and $S_{100}$





was found (Fig. 7). This weak correlation is responsible for the better agreement obtained with $\varepsilon_S$ than with $\varepsilon_N$. More studies
are necessary to analyze the dependence between $\sigma$ and $S_{100}$ under more suitable conditions (i.e. less variable $L_{out}$ and low $Ri$).

(6) Reciprocally, the poorer statistical agreement between $\varepsilon_N$ and $\varepsilon_U$ leading to a regression slope less than 2/3 (0.50, almost identical, 0.55, that obtained by L18 from MU radar data) reveals that $\sigma$ has in practice a statistical degree of dependence with $1/N$ as confirmed by Fig. 4.

(7) The combination of (6) and (7) leads to the conclusion that, to some extent, $\sigma$ depends on $Ri_{100}^{-1/2}$ at least for $Ri_{100} > \sim 0.4$ (Fig. 8). This dependence does not seem to be valid for lower $Ri_{100}$ (Fig 8a), in accordance with the fact that N should not affect turbulence when the Richardson number is low (Eq. 4).

(8) The analysis of the three models $\varepsilon_{Lout}, \varepsilon_N$ and $\varepsilon_S$ with $\varepsilon_U$ vs $Ri_{100}$ (Fig. 9) confirms the good agreement between $(\varepsilon_{Lout}, \varepsilon_U)$ and between $(\varepsilon_S, \varepsilon_U)$ and the inadequacy of $\varepsilon_N$ for $Ri_{100} \lesssim 1$. The underestimation of $\varepsilon_N$ increases as $Ri_{100}$ decreases. The results for $Ri_{100} \gtrsim 1$ are more difficult to interpret and more puzzling, but $\varepsilon_S$ and $\varepsilon_{Lout}$ lead to comparable results and do not show substantial bias as a function of $Ri_{100}$. In any case, all results involving large $Ri$ (> 1) must be taken with caution, because the turbulence may be intermittent. In principle, the interpretation of the results should consider this.

(9) We compared TKE dissipation rates obtained from the Thorpe analysis of simultaneous radiosonde data with DataHawk and radar estimates for two turbulent layers (Fig. 2). We tested two models. The classical model $\varepsilon_T = c^2 L_T^2 N^3$ (Eq. 5) based on the equivalence between the Thorpe length $L_T$ and the Ozmidov scale $L_O$ ($c = 1$) fails to reproduce DataHawk-derived $\varepsilon$ in the K-H layer for which $Ri$ is expected to be less than 0.25. Although the disagreement can be due to several factors (e.g. an inappropriate choice of $c$, horizontal inhomogeneity), it can also be due to the fact the model involves the Ozmidov scale defined for a turbulence affected by the stable stratification. In essence, $L_T$ cannot be related to $L_O$ anymore by a constant if the stratification effects can really be neglected for low $Ri$. Therefore, an alternative approach using the Corrsin scale $L_C$ instead of $L_O$ was introduced, leading to $\varepsilon_T' = c'^2 L_T^2 S^3$ (Eq. 7), compatible with studies showing a $Ri^{3/4}$ dependence of $L_T/L_O$ for $Ri < 0.25$. Contrary to $c$, $c'$ is a true constant (with respect to $Ri$) for low $Ri$. It is worth noting that Eq. (7) and Eq. (3) form a coherent pair of models independent of $N$ for a weak stratification or strongly sheared flows. Using values of $c'$ deduced from the literature, $\varepsilon_T'$ provides estimates consistent with DataHawk and radar-derived $\varepsilon$ (expect $\varepsilon_N$) for the K-H layer. On the other hand, $\varepsilon_T'$ fails for a decaying turbulent layer ($Ri > 1$) as the model is not expected to be valid for $Ri > 0.25$. $\varepsilon_T$ with $c = 1$ shows a better agreement with DataHawk and radar-derived $\varepsilon$ (including $\varepsilon_N$), coherent with the fact that the stable stratification should affect the turbulence for large $Ri$. These results need to be confirmed by statistical studies.





**Appendix 1.**

The turbulent Froude number $Fr = \varepsilon/Nk$ where $k$ refers to TKE for a simple and standard notation is often used to characterize turbulent mixing (e.g. Ivey and Imberger, 1991). A strong and weak stratification is associated with $Fr < 1$ and $Fr > 1$, respectively. $T_L = k/\varepsilon$ is called the inertial time scale and is a characteristic time of TKE dissipation. The corresponding time scales associated with the turbulence production by the wind shear and with the conversion into potential energy are $S^{-1}$ and $N^{-1}$, respectively. When the stratification is weak, i.e., when $NT_L = Fr^{-1} < 1$, then $T_L$ (dissipation time)

and $S^{-1}$ (production time) should be of the same order, i.e. the shear parameter $ST_L = O(1)$, for stationary turbulence. Several studies (e.g. Mater and Venayagamoorthy, 2014 and references therein) reported a critical value for weakly stratified and stationary flows:

$$ST_{Lc} \approx 3.33 \tag{A1.1}$$

By dividing Eq. (A1) by 3.33 $NT_{Lc}$, we obtain:

$0.3\, S/N = 1/NT_{Lc}\ (\Leftrightarrow Fr = 0.3/\sqrt{Ri})$       (A1.2)

From the definition of $Fr$, Eq. (A1.2) reads:

$$\varepsilon = 0.30\, k\, S \tag{A1.3}$$

Assuming isotropy, $k = 3/2\, \langle w'^2 \rangle$., where $\langle w'^2 \rangle$ is the vertical velocity variance (assumed to be $\sigma^2$ in the paper). We obtain:

$$\varepsilon = 0.45\langle w'^2 \rangle\, S \tag{A1.4}$$

For (anisotropic) shear generated turbulence, $k \approx 2\langle w'^2 \rangle$, so that

$$\varepsilon = 0.60\langle w'^2 \rangle\, S \tag{A1.5}$$

i.e., virtually Eq. (3) with $C_s = 0.63$. These expressions are valid for $Fr > 1$, i.e. $Ri < 0.09$, according to (A1.2).

For $k \approx 2.74\langle w'^2 \rangle$ when $0 < Ri < 0.2$ (Eq. (28), Basu and Holtslag (2021)), we get:

$$\varepsilon = 0.82\langle w'^2 \rangle\, S \tag{A1.5}$$


**Appendix 2.**

From Fig.(3) of Garanaik and Venayagomoorthy (2019) showing the turbulent Froude number $Fr$ vs $L_E/L_O$ (or $L_T/L_O$) from DNS, we obtain for weakly stratified conditions ($Fr > 1$):

$Fr = \alpha(L_T/L_O)^{-2/3}$       (A2.1)

with $\alpha \approx 0.7$. Note that this coefficient is deduced from their Fig. (3) using the linear trend shown by the authors. They did not explicitly refer to this value. By using the definitions $Fr = \varepsilon/Nk$ (see Appendix 1) and $L_O = \sqrt{\varepsilon/N^3}$, (A2.1) can be re-written as:

$$\varepsilon = (Fr/\alpha)^3\, L_T^2 N^3 \tag{A2.2}$$

For $Fr > 1$ or $NT_L = Fr^{-1} < 1$, $ST_L \approx 3.33$ (see Appendix 1 and Fig. (1) of Mater and Venayagamoorthy, 2014). Therefore, $Fr \approx 0.3\, S/N$ so that:

$$\varepsilon = (1/\alpha)^3\, L_T^2 S^3 = c'^2 L_T^2 S^3 \tag{A2.3}$$

with $c' = 0.28$. This value agrees well with those reported in the main text.



In addition, by replacing $Fr$ by its definition, Eq. (A2.2) can be re-written as $\varepsilon = \alpha^{-3} L_T^2 \varepsilon^3 / k^3$ so that:

$$\varepsilon \approx k^{3/2} / (1.5 L_T) \tag{A2.4}$$

Eq. (A2.4) provides a way to relate the isotropic turbulent length scale $L_k$ defined as the scale of the largest eddies weakly affected by the buoyancy and the shear to the Thorpe length: $L_k \sim 1.5\, L_T$. It also provides an expression of the master length scale $L_M$ defined as $(2k)^{3/2} / B_1 \varepsilon$ (e.g. Mellor and Yamada, 1982) with $11.9 \leq B_1 \leq 27.4$ (Table 2, Basu and Holtslag, 2021).

We obtain $L_M = 4.24 / B_1\, L_T$ .

On the other hand, for strongly stratified conditions ($Fr < 1$), we can write, from Fig. (3) of Garanaik and Venayagomoorthy (2019):

$$Fr = \alpha' (L_T / L_O)^{-2} \tag{A2.5}$$

with $\alpha' \approx 0.6$ according to the figure, or, purposely, $0.66 = 2/3$. We obtain:

$\varepsilon = 3/2\, Fr\, L_T^2 N^3$

so that:

$$k = 3/2\, L_T^2 N^2 \tag{A2.6}$$

For $k = 3/2 \langle w'^2 \rangle$, then, $\langle w'^2 \rangle = L_T^2 N^2$ or:

$$L_B = \sqrt{\langle w'^2 \rangle} / N = L_T \tag{A2.7}$$

If $k = \beta \langle w'^2 \rangle$, then $L_B = \sqrt{1/(\alpha'\beta)}\, L_T$. Eq. (A2.5) demonstrates that the equivalence (or least the proportionality) between the buoyancy scale and the Thorpe length is valid for strongly stratified conditions only (i.e. for conditions for which stability affects the vertical motions before being affected by the wind shear). For a weak stratification, the vertical TKE cannot be fully converted into potential energy because the parcels cannot move vertically over a length of $L_B$ (but $L_H$ only). Therefore, the basic stability does not intervene anymore in the variance of the vertical velocity fluctuations as in the case of a neutral
stratification.

*Data availability*. The WPR-LQ-7 data are available on http://www.rish.kyoto-u.ac.jp/radar-group/blr/shigaraki/data/

*Author contributions*. HL wrote the paper with the help of LK, HH, DL, AD, TM and MY for the conception of the study, the collection, post-processing of the radar and UAV data and retrievals.

*Competing interests*. The authors declare that they have no competing interests.

*Financial support*. This study was partially supported by JSPS KAKENHI Grant JP15K13568 and the research grant for Mission Research on Sustainable Humanosphere from Research Institute for Sustainable Humanosphere (RISH), Kyoto University.




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
