# Peer review of "Turbulence Kinetic Energy dissipation rate: Assessment of radar models from comparisons between 1.3 GHz WPR and DataHawk UAV measurements"

_Atmospheric Measurement Techniques, 2023_

## Author Comment (AC1)

**Reply to Referee 1**

This paper combines a nice concise review of the current state of play with measurement of turbulent energy dissipation rates in the atmosphere by in-situ and radar methods. It is backed up by some good experimental data as well, and contains nice side-by-side radar/in-situ comparisons.

I applaud the effort, and will recommend publication - but subject to a few caveats.

We thank the referee for his kind comments and encouragement.

First, there is some lack of clarity in the terms TKE and ∈. The abstract seems to suggest that they are almost the same, but one is a total energy and one is a dissipation rate. But in line 120 and elsewhere they are treated as different entities - maybe I am misinterpreting something, but I think TKE and e need to be more carefully defined.

We do not clearly understand this reviewer comment because TKE (alone) is not defined in the abstract (we introduced its dissipation rate $\varepsilon$ only). However, we replaced "TKE dissipation rate" by "dissipation rate of turbulence kinetic energy (TKE)" to eliminate any confusion.

As an introduction to the equations recalled in line 102 for the steady state and in line (120) for the unsteady state, a simplified equation of TKE budget equation is now introduced in the first paragraph of Introduction as follows:

"The dissipation rate $\varepsilon$ ($\mathrm{m^2 s^{-3}}$ or $\mathrm{mWkg^{-1}}$) of turbulence kinetic energy (TKE) is an important variable for assessing heat deposition by turbulence in the atmosphere. This variable appears in a simplified expression of the ensemble-mean TKE budget equation (see, e.g., Stull (1988) for a complete derivation and for its conditions of validity):

$$\partial TKE / \partial t = P - B - \varepsilon \tag{1}$$

where $P$ is the shear production term and $B$ the buoyancy flux term."

A more serious error occurs in lines 136 to 142 - and especially line 142, where it says

*"In essence, there is no contribution from an anisotropic buoyancy subrange."* The theory of Hocking (1983) does NOT assume this. The factor of 1/2 in Hocking (1983), eq, (2) comes from a crude assumption that the radar receives half of it's v^2_rms from the inertial range and half from the buoyancy range. (In fairness to the readers, this was not fully explained in the original paper.)

I would ask the authors to please look at Hocking et al. (2016) [Atmospheric Radar: Application and Science of MST Radars in the Earth's Mesosphere, Stratosphere, Troposphere, and weakly ionized regions", Cambridge University Press, 2016. ISBN 9781316556115, DOI: https://doi.org/10.1017/9781316556115], pages 407-408.

The more advanced theory presented there allows for a variable contribution to v_rms from BOTH the inertial range and the buoyancy range.

So the theory ascribed to equation (2) of the current paper is not the latest - **in a more complete form, it should include this factor F - see the pages indicated above.**

Indeed it might be possible that this fraction F could be allowed to be dependent on $\in$, which could allow better consolidation between the theories presented in the paper under review. Physically, this is not unreasonable - for example, more intense turbulence could more rapidly destroy the stratification, allowing for larger isotropy at larger scales - even into the buoyancy range -- and this would allow a larger contribution to the vertical RMS velocities from the buoyancy range, reducing the fraction F.

Indeed the behaviour of this "buoyancy range" is not fully understood at all - it is clear that turbulence here is anisotropic, with suppressed vertical velocities relative to the horizontal ones, but exactly what this ratio is is quite unclear. It also leads to huge levels of confusion relating to the so-called "buoyancy scale" of Weinstock and the similar but different" Ozmidov scale".

So I ask the authors to at least refer to this later work, and incorporate this fraction F into discussions. They may choose to say that "for a fixed value of F" it does not agree with the eq (1), but it might reconcile better if F varies with $\in$, and might allow further insight into why eq. (1) seems to work so well (which is still a bit of a mystery, as discussed by the authors in section 5.1).

*And*

Page 15 - various limitations of $\in$_N are discussed, but remember these all assume a fixed value of F, so it must be made clear that this is the simplest version of this model, not the most complete.

We agree with the above comments of the reviewer. We are very familiar with the seminal treatise on "Atmospheric Radar" by Hocking et al. (2016) and we do reference it. In any case, in response to these comments, we have displaced the sentence *"In essence, there is no contribution from an anisotropic buoyancy subrange"* to a more suitable place in the text which referred to the model for weak or strongly sheared stratification. We also modified the paragraph following Eq. (2) (now Eq. (3)) as follows:

"…where $C_N$ is a constant. This expression is expected to be valid for turbulence in a stable stratification ($N^2 > 0$) whose an outer scale is defined by the buoyancy scale expressed as $L_B = \sqrt{\langle w'^2 \rangle}/N = \sigma/N$. Eq. (3) is thus equivalent to $\varepsilon_N = C_N \sigma^3 / L_B$. In a pioneering contribution, Hocking (1983) first derived Eq. (3) from the integration of the transverse 1-D spectrum of vertical velocities over the inertial and buoyancy subranges to relate $\varepsilon$ to $\langle w'^2 \rangle$. In its original derivation, the author assumed roughly equal contributions to $\langle w'^2 \rangle$ from the inertial and buoyancy subranges. More recently, Hocking et al (2016) proposed a more general expression by introducing a variable factor F, where F is the ratio of the buoyancy contribution to the inertial subrange contribution. This factor can vary from 0.5 to 1. It affects the value of the constant $C_N$ and Hocking et al. (2016) recommends that a value of ($0.5 \pm 0.25$) be used, which takes into account the variability of F, difficult to determine in practice."

Indeed (jumping ahead a bit!) I found section (5.1) quite unhelpful in this regard, and I sense that the authors have similar issues, so it might be useful here to consider further the relative roles of vertical velocities in the buoyancy and inertial ranges, which relates in turn to F.

(remembering that most of the issues here arise because the pulse-length and beam widths are right around the buoyancy scale)

*And*

Section 5.1 - see my earlier comment.

See our reply above.

Another point to bear in mind in regard to section 5.1 relates to Figs. 7.12 and 7.13 of Hocking et al. (2016). While it is easy to believe that the Datahawk is somehow more "perfect" than a radar, as it samples at a single point in space, determination of a spectrum (or alternatively an autocovariance) function requires a finite length of time, and in that time, the datahawk moves, and additionally the mean wind blows different regions of space across the datahawk as well, so the datahawk also has a spatial sampling across many tens of metres, just like the radar. This may also relate to the "70m" scale.

The reviewer is right, and we recognize that considering UAV-derived $\varepsilon$ as an "absolute" reference is not fully justified for the advocated reasons: The "70-m" scale may be dependent on the UAV measurement method. For this reason, the empirical model with Lout=70 m must be tested (as well as $\varepsilon_S$ and $\varepsilon_N$) independently from UAV measurements, in order to determine the representativeness of this value. This can be made by comparing the results of the models to other data from other instruments (for example, a Doppler lidar) and for other cases of turbulence exhibiting well-developed KH billows (for which Ri values are expected to be much smaller than 0.25). These results will be presented in future papers.

Returning now to section 2, the discussion in section 2.2 is great.

However, I do note some inconsistencies in the text as to the terms Ri and Rf - in places they are written as subscripts (R_f) and sometimes not. Please decide which is best.

We made the necessary changes.

Section 3 is straightforward, though the authors refer to 59-s data sets and 1-min data sets, which I assume are one and the same (??).

Yes, "1-min" is an approximation. The correct value is 59 s.

Section 3.3 -- Ozmidov scales and Thorpe lengths are discussed, but they also relate to Weinstock's "Buoyancy scale" and there is a factor of 10 ($2\pi/0.6$) difference here - may not be relevant in section 3.2 but certainly relevant in a general context.

We did not consider the 2*pi factor in our work as is often the case in fluid mechanics literature.

Line 210 - please more formally define "TKE" and distinguish it from e - maybe it was done earlier(?) - if I missed it, I apologize - but I think the distinction needs to be clear.

See our first response.

The case studies (section 4) seem well documented.

Thank you.

Figs 5 and 6 - note that references to the $\in$_**N** formula assumes F=0.5 (see earlier)

It is mentioned in the revised paragraphs

Fig 6 - please add values of R and P, as you have done for Fig. 7

Done. Note that the correlation coefficients are high due to self-correlation (but variables contain $\sigma^2$).

Line 392 - Terns like "cst" appear -  I am not clear what cst means!?  I could not find a definition.  Please clarify.

"cst" means "constant". This abbreviation is not used in English. Sorry for the misunderstanding.

**Conclusion** - great paper, nice review, nice data, but please recognize that the latest version of the model for Eq. (2) has NOT been used, and the authors have assumed a fixed fraction F (relative contributions of inertial and buoyancy scales). If the authors continue using the current discussions, they are obliged to recognize verbally that they have assumed a simplified model with fixed F, which may not be realistic. Using the 1983 version of the theory pertaining to Eq (2) and not considering the 2016 version is a bit unfair.

The corrections have been made (please see above).

The case of the radar  resolution being much less than the buoyancy scale has also been discussed in Hocking et al. (2016) pages 409-411, and of course also in the paper by Kantha et al., (2017),and all parties agree that in such cases $\in$ should be proportional to vrms^3 - it should be noted that this is true, and the complications discussed above arise from the cases where the pulse -length exceeds the buoyancy scale (or equivalent) and where data-lengths are not too long.

From our point of view, the statistical $\sigma^3$ law cannot be explained by the sole condition of the radar resolution much less than the buoyancy scale. It may be the case on some occasions but not always (as we showed for one case in the manuscript, for example, but there are many other cases as described by Luce et al. 2018). We speculate that this observation is the result of a multitude of different situations depending on multiple factors (sources and stage of turbulence, stability, shear, volume, etc.) so that a given model (e.g. $\varepsilon_N$ or $\varepsilon_S$) can be valid but for specific conditions and may not be applicable to individual cases, but is satisfactory as a statistical average over an ensemble of various conditions..

One positive point from this paper is that it seems we as a community are getting better at measuring $\in$ values - contrast this to Fig.1 of Hocking and Mu, in which even the definitions of "light", "moderate" and "heavy" turbulence differed by factors of 10 and more, depending on the author. Despite all the concerns in this paper, Fig. 6 shows that between $\in$ = 10^{-4} and 10^{-2}, all models agrees to better than half a decade (factor or 3) nowadays.

Thank you. Once again, we thank this reviewer for a thoughtful review.

---

## Author Comment (AC2)

**Reply to Referee 2**

**Turbulence Kinetic Energy dissipation rate: Assessment of radar models from comparisons between 1.3 GHz WPR and DataHawk UAV measurements**

By Luce et al.

**Summary of paper:**

This paper presents some derivations of turbulent energy dissipation rates epsilon in the lower atmosphere with in-situ and radar techniques. The authors give a nice review of the algorithms to estimate epsilon mostly used in the community, although some of them have been assessed and some are used with different hypotheses. Based on the analysis of dataset on a campaign-basis, they further estimate epsilon and compare the results derived from different techniques for case studies as well as for statistical analysis.

**General comments:**

The research topic of turbulence detection in the atmosphere is very interesting and important and certainly relevant to the readers of AMT. The authors have a unique dataset with in-situ and remote sensing instruments at their disposal that are certainly valuable for the corresponding research. The manuscript is well written and organized and of course deserves to be published.

We thank this reviewer for his positive and constructive comments.

However, there are several points should be clarified before publication.

We have tried to answer them below.

Different probing techniques have different spatial coverage: in-situ instrument for a point and radar for the cross section of a volume. The necessary discussions or significance test should be given when one compares the derived epsilon from different techniques. Otherwise, the resulting consistency seems to be coincident, especially for the case studies.

We agree with the reviewer, but this pertinent comment applies to all the numerous studies published in the literature comparing remote sensing (volume-averaged) and in-situ (point) measurements. However, we are aware that the different characteristics of the two measurements is an important source of uncertainties. This is one of those problems, which cannot be realistically addressed, since the resources needed to do so are unrealistic. We expect the reader to be aware of what we are doing and its limitations.

For any study of turbulence intensity estimate, it is very important to describe the details of each steps. Each technique suffers from different instrument limitation. The error bars for the measurements are very important and may play a determining role in the results. The discussions on the error sources for each instrument, in my opinion, are very important and necessary.

We understand the reviewer comment and it is partly due to an unfortunate error in the reference list. The reference that explains in detail the processing of UAV and radar data is incorrect and should have been:

Luce, H., Kantha, L., Hashiguchi, H., Lawrence, D., and Doddi A.: Turbulence Kinetic Energy Dissipation Rates Estimated from Concurrent UAV and MU Radar Measurements, Earth Planets Sci., 70: 207, 2018.

Instead, Luce et al. (2018, PEPS), which is not related to the present work was cited. Another paper (*H. Luce,* L. Kantha, H. Hashiguchi, and D. A. Lawrence, Estimation of Turbulence Parameters in the Lower Troposphere from ShUREX (2016-2017) UAV Data, **Atmosphere**, 10, 384-409, 2019) describes in detail the data processing of the UAV measurements.

Relevant corrections to the reference list and citations have been made.

The manuscript is thus the sequel of several previously published works and similar approaches have been applied by Dehghan et al. (2014) (comparisons radar – aircraft).

Discussion about the possible sources of uncertainties:

Dehghan and Hocking (2011) enumerated various uncertainties in estimating various uncertainties in radar measurements of turbulence. This reference has been added and cited in the revision. A synopsis is given below for the reviewer.

(1) The radar measurements are volume- and time-averaged. The volume average plays the role of a spatial filtering removing the contribution of eddies with the largest dimensions if their size exceeds the dimensions of the radar volume. The radar method consists in estimating the variance $< w'^2 >$ of the vertical wind disturbances produced by turbulence (when the radar beam is vertical). The radar estimate $\sigma^2$ is an unbiased value of $< w'^2 >$ if all the scales contribute to the radar signals. The validity of this hypothesis is a priori unknown and is a first source of uncertainty (impossible to estimate). It is all the more difficult to estimate as the measurements are not instantaneous and are obtained over the time required to obtain a Doppler spectrum (24 s for the MU radar and 1 minute, intermittently, for the LQ7 wind profiler). The (potential) spatial filtering effects are thus partially counterbalanced by the advection of the turbulent eddies during the acquisition time. On the other hand, the time acquisition must be sufficiently small to reduce other possible (atmospheric) sources of vertical wind fluctuations such as those produced by the full spectrum of gravity waves and the decorrelation time associated with the decay of the turbulent eddies. The radar configuration used should be adequate for a minimization of these effects, but they cannot be totally avoided. They are another source of errors, likely extremely variable, when equating $\sigma^2$ to $< w'^2 >$.

(2) Another source of uncertainty is due to the hypothesis that turbulence fills the entire radar volume and is homogeneous. This hypothesis is required to remove the non-turbulent contributions due to beam- and shear- broadening (see, e.g. Dehghan and Hocking, 2011, Nastrom, 1997). The procedure was recalled by Luce et al. (2018) and is not explicitly introduced in the manuscript because it is widely used in the field:

$$\sigma^2_{(fluct)} = \sigma^2_{(obs)} - \sigma^2_{(non-turb)}$$

Where $\sigma^2_{(obs)}$ is the measured variance, $\sigma^2_{(non-turb)}$, the corresponding variance due to non-turbulent effects, and $\sigma^2_{(fluct)}$ the variance supposed to be due to turbulence ($\sigma^2$ noted above).

For measurements made at vertical incidence, $\sigma^2_{(non-turb)}$ is proportional to the square of the mean wind speed U at the altitude of the radar gate. U is also affected by uncertainties related to the method used to estimate it. U is derived from the measurements of radial winds in 4 oblique beam directions, assuming that the horizontal wind is uniform over a horizontal distance corresponding to the separation of the radar volumes, and stationary during the acquisition time. Any departure from these assumptions is a source of uncertainty. In addition, there is a debate on the effective horizontal wind to use. The "instantaneous" wind speed (i.e. the wind measured at the same time resolution as $\sigma^2_{(obs)}$) may not be representative of the background wind to consider (because itself affected by turbulence and waves), and wind speeds averaged over much longer periods (e.g. hourly winds) can be used instead (e.g., Dehghan and Hocking, 2011). It turns out that this alternative may produce substantial differences, especially when the wind changes quickly such as in frontal zones for example.

(3) Third source of uncertainty is obviously the estimation error of the spectral width which primarily depends on $\tau$ , the length of the time series (and thus, on the spectral resolution $1/\tau$). A statistical estimation error is given by (19) of Dehghan and Hocking (2011):

$$\delta\sigma^2_{(obs)} = \lambda\sigma_{(obs)}\eta/\sqrt{2log2}\tau$$

where $\lambda$ is the radar wavelength and $\eta$ is a parameter varying between 1 and 1.7. Assuming that $\sigma^2_{(obs)}$ is assimilated to $\sigma^2_{(non\_turb)}$ (to establish an order of magnitude), then we get for the LQ7 WPR, $\delta\sigma^2_{(obs)} \sim 10^{-4} - 10^{-3}~m^2 s^{-2}$ for typical horizontal winds met during the campaign (5-10 m/s). For $\varepsilon_{Lout}$ model, it corresponds to an estimation error of $\delta\varepsilon_{Lout} \sim 1 - 5~10^{-7} m^2 s^{-3}$, i.e. much less than the estimated levels (typically $10^{-6} - 10^{-2}~m^2 s^{-3}$). Therefore, we can conclude that the estimation errors are much less than all the other possible (combined) sources of uncertainties.

(4) Finally, (3) assumes that the estimates are not affected by noise (which is the case for large signal to noise ratios and Gaussian fitting method) and not affected by non-atmospheric echoes (outliers, such as birds, insects, clutter, airplanes…and UAV in the present case). These possible artefacts have been removed by automatic procedures for the LQ7 WPR and manually removed for the MU radar, but residual contributions can never be excluded. Some spectra have been rejected when the contaminations are too important so that the averaged profiles do not always correspond to the average of all profiles collected during the 10, 20 or 30 minutes. This is another source of variability.

The above discussion is likely not exhaustive and indicates that the estimation of confidence intervals is difficult or even impossible considering all the necessary hypotheses and steps used to obtain the final products, not to mention all the uncertainties associated with N (BV frequency) dependence for $\varepsilon_N$ (N is estimated from balloon or UAV data) and with S (shear) dependence for $\varepsilon_S$. That's why the literature on the topic in general does not give this kind of information. The statistics from multiple events are the only way to reveal significant trends.

In the present case, we recognize that the agreements for the case studies can be a priori fortuitous (it is the weak point of the absence of error bars). However, we show the comparisons between 4 profiles of $\varepsilon_{UAV}$ and the profiles from 2 different radars (the MU radar and LQ7 wind profiler). The mean values of all the UAV- and radar-derived estimates (except $\varepsilon_N$ because the model is expected not to be valid) show very close levels with the same temporal variations

(Figure 3). This agreement should reveal a significant trend and would have been much more debatable if we had only one UAV profile compared to a single radar profile.

In the revised version, we temper the impact of our results for the case studies by not ruling out some degree of coincidence. However, they represent a serious clue, taking into account all sources of error and uncertainty. The conclusions of statistical analyses carried out with the LQ7 WPR are less affected by these uncertainties, as they result from the combination of numerous comparisons. They also reveal trends identical to those observed with the MU radar (Luce et al. 2018) for $\varepsilon_N$ and $\varepsilon_{Lout}$, confirming our analysis.

One of the messages of our manuscript is the equivalence of $\varepsilon_S$ and $\varepsilon_{Lout}$ for the KH layer, allowing us to establish a possible interpretation of the value of $L_{out} \sim 70$ m (found to be a canonical statistical overall average from both MU and LQ7 radars) for such a situation. The quantitative agreement between the two may be a coincidence. In particular, we believe that the depth of the KH layer was, by chance, suitable for such a good agreement. In a near future, we will present an analysis of radar data in support of the validity of our conclusions.

It is mentioned in the manuscript that the value of c (i.e. the ratio of L_O to L_T) is tricky by applying the Thorpe analysis. Different authors used different c values for their calculations. This is because the experimental validations for this ratio in the atmosphere are very sparse. Based on several balloon flights, Schneider et al. (2015) directly checked the relation of L_O and L_T and they found the distribution of c^2 covers a very broad range of 2 orders of magnitude. In my opinion, this work (https://doi.org/10.5194/acp-15-2159-2015) is worthy to be referenced in the current manuscript.

We do not include this reference because later work by the same group showed that the balloon data were corrupted up to about 60 % of the time (Söder et al., 2019, Evaluation of wake influence on high-resolution balloon-sonde measurements, AMT, https://doi.org/10.5194/amt-12-4191-2019). In their own words, the results were qualified as "questionable", explaining the unrealistic levels of turbulence for large Ri's i.e. for weak shears (and thus enhanced probability of balloon wake).

In this manuscript, the authors give a nice review of the different algorithms to estimate epsilon with a large number of symbols. For ease of reading, I recommend the authors to add a list of symbols (maybe also including abbreviations).

We included a list of symbols

**Typos and suggestions listed as follows (but not limited to):**

1. *Line 14, does "ASL" mean above sea level? It should be clarified elsewhere.*

   Yes, we clarified this point.

2. *Line 60, N is B-V frequency, not N^2.*

   It is corrected.

3. *Line 79 and line 84, the references should be accurate (Hocking, 2016 and Hocking et al., 2016).*

   It is corrected: Hocking et al. (2016)

4. *Line 79, the sentence "… Appendix of L18" is misleading. Please check.*

   Unfortunately, L18 was not correctly referenced in the reference list. We apologize for this unfortunate error because it was an important reference for understanding the context of our manuscript.

5. Line 109, what do the arrows mean here? (And the arrows used elsewhere) Please clarify them or describe in text.

   The arrow means "goes to" as for the calculus symbol (e.g. "x goes to 0", "$x \to 0$"). We think that it is very convenient and avoids unnecessary text.

6. *Line 122, please reconsider the subscripts, like Ri_S and Rf or R_f elsewhere.*

   It is corrected. We used the notation $R_f$

7. *Line 123, epsilon = P/G – B;*

   It is corrected.

8. Lines 140-141, please add references?

   There is no dedicated reference, but a similar comment can be found in Doviak and Zrnic', Doppler radar and weather observations, Academic Press, San Diego, 1984, page 338 , 10.52-10.53).

9. *Line 156, does AGL mean above ground level? Why do the authors use both ASL and AGL?*

   Yes. Different datasets often use different reference levels (AGL or ASL).

10. Line 158-159, please consider to add any reference for the processes or algorithms to remove outliers?

    Algorithms similar to those proposed by  B. L. Weber and D. B. Wuertz, Quality Control Algorithm for Profiler Measurements of Winds and Temperatures, U.S. Department of Commerce, National Oceanic and Atmospheric Administration, Environmental Research Laboratories, Wave Propagation Laboratory, 1991) were applied.

11. Line 267, Figure 12 not found. Please be sure.

    L18 was not correctly referenced in the reference list. We apologize again.

12. *Line 320, please remove "4.1.3 Comparison with epsilon_T"*

It is corrected.

13. Lines 335-336, please rephrase the sentence (Depending on …).

It is rephrased as follows: "Depending on the flight segment (A1, D1, A2, D2), $c = \langle L_O \rangle / L_T$ was found between 0.65 and 1.56 and ~1 in average."

14. Line 362, please rephrase the sentence with more clear expression.

It is rephrased as follows: From Eq. (7) with $Ri = 2$, $c'_{Sc} = 0.41$ and $c'_{BP} = 0.24$, we obtain $\varepsilon'_T \approx 0.012 \ mWkg^{-1}$ and $0.004 \ mWkg^{-1}$, respectively which is much less than $0.2 mWkg^{-1}$

15. Line 392, what does "cst" mean? Constant? It is also used elsewhere. Please clarify.

"cst" means "constant". It is corrected. This abbreviation is not used in English. Sorry for the misunderstanding.

16. Line 426, what is P test? Please add references!

The P test (or P value) refers to a standard evaluation of the statistical significance of the correlation coefficient. This parameter is automatically evaluated by, e.g., Matlab. Please see the documentation help from Matlab or information here:

https://fr.mathworks.com/matlabcentral/answers/141942-corrcoef-p-value-interpretation

Once again, we thank this reviewer for a thoughtful review.